# Online Learning in the Repeated Mediated Newsvendor Problem

**Nataša Bolić**[1] **Tommaso Cesari**[1] **Roberto Colomboni**[2,3] **Christian Paravalos**[1]

[1]EECS, University of Ottawa, Ottawa, Canada
[2]DEIB, Politecnico di Milano, Milano, Italy
[3]Department of CS, Università degli Studi di Milano, Milano, Italy
`{nboli039,tcesari,cpara006}@uottawa.ca` `roberto.colomboni@polimi.it`

## Abstract

Motivated by real-life supply chain management, we study a repeated newsvendor problem in which the learner is a mediator that facilitates trades between suppliers and retailers in a sequence of supplier/retailer interactions. At each time step, a new supplier and retailer join the mediator's platform with a private production cost and utility function, respectively, and the platform proposes a unitary trading price. The supplier accepts the proposed price if it meets or exceeds their unitary production cost and communicates their decision to the platform; simultaneously, the retailer decides the quantity to purchase at the proposed trading price based on their private utility function and sends their decision to the platform. If the supplier accepts the trading price, the transaction proceeds, and the retailer purchases their chosen quantity of units, paying the product of this quantity and the trading price to the supplier. The mediator's objective is to maximize social welfare. We design an online mediator's pricing strategy that features sharp regret rates under some natural assumptions, and we investigate the necessity of these assumptions, proving that relaxing any of them leads to unlearnability.

## 1 Introduction

The newsvendor problem is a central topic in the analysis of supply chains. In its classical one-shot version Arrow et al. [1951], the supplier chooses and reveals a trading price. After observing the trading price, the retailer orders a certain quantity of the good, deciding how much to order based on a privately held utility function that depends on their needs and wants. The repeated version studied in Huang and Sošić [2010] and Cesa-Bianchi et al. [2023a] generalizes this problem to a sequence of supplier/retailer interactions.

### 1.1 Motivations

To the best of our knowledge, the newsvendor problem has only been studied under the assumption that the supplier determines the trading price. This assumption falls short in *many* real-life applications where an independent, third-party mediator, whose goal is to serve the interests of both parties, selects the trading prices. The following examples illustrate markets where mediator-based pricing could naturally arise and prove to be beneficial. *Digital advertising intermediaries* (e.g., ad-exchange platforms) propose clearing prices for ad impressions or clicks to publishers, and advertisers decide how many to purchase. *Cloud computing platforms* (e.g., AWS, Azure, GCP) propose pay-as-you-go rates to data centers or resource providers, and users needing computing power decide how much of it to use. *Real estate rental intermediaries* (e.g., Airbnb for short-term accommodations or corporate housing for long-term stays) propose nightly, weekly, or monthly rates to property owners, and renters decide the duration of their stay. *Online freelance marketplaces* (e.g., Upwork, Fiverr) propose hourly

or per-task rates to freelancers (e.g., designers, programmers), and users decide how many hours or tasks to book. *Ridesharing platforms* (e.g., Uber, Lyft) select the per-mile prices of rides, drivers decide to accept or reject the prices posted by the platform based on the cost of fuel/car maintenance, and riders choose the distance of the ride. Lastly, *microgrid energy operators* propose real-time prices to households capable of generating energy, and other households or businesses choose how much energy to purchase.

In addition to practical motivations, this problem is also intriguing from a theoretical point of view. Indeed, the repeated newsvendor problem is a strict generalization of the popular *dynamic pricing* problem. Consequently, our proposed repeated *mediated* newsvendor problem strictly generalizes *bilateral trade* (see Appendix A for a formal proof of this claim), a setting that has recently garnered significant interest within the machine learning community [Cesa-Bianchi et al., 2021, 2023b,c, 2024, Azar et al., 2022, 2024, Bernasconi et al., 2024, Bolić et al., 2024, Bachoc et al., 2024, Gaucher et al., 2025, Hajiaghayi et al., 2025, Deng et al., 2025]. Notably, our tight regret guarantees also recover optimal regret rates (up to logarithmic terms) in the important special case of bilateral trade.

All of these considerations give compelling practical and theoretical reasons to study this problem and characterize its learnability.

## 1.2 Formal setting

Let USC be the set of all $[0,1]$-valued upper semi-continuous functions defined on $[0,1]$. Select, for any $u \in$ USC and price $p \in [0,1]$, $u^{\#}(p) \in \operatorname{argmax}_{q \in [0,1]} \big( u(q) - p \cdot q \big)$ (our theory holds irrespectively of how this selection is made, i.e., of how ties are broken) and define $u^{*}(p) := u\big(u^{\#}(p)\big) - p \cdot u^{\#}(p)$. Up to a simple rescaling, we can and do assume without loss of generality that trading prices, suppliers' costs, and quantities exchanged belong to $[0,1]$. We also assume that retailers' utilities belong to USC, a minimal condition needed for $u^{\#}$ to be well-defined.[1]

The online learning protocol for the repeated mediated newsvendor problem (RMNP) we study proceeds as follows.

---

**Online learning protocol**

---

1: **for** each time $t = 1, 2, \ldots$ **do**
2:   A supplier with a private[2] cost $C_t$ and a retailer with a private[2] (gross) utility $U_t$ join the mediator
3:   The mediator proposes a trading price $P_t$
4:   The supplier communicates $\mathbb{I}\{C_t \leq P_t\}$ (i.e., whether they accept the trade, which happens if and only if the proposed trading price $P_t$ is no lower than their production cost $C_t$), and the retailer communicates $U_t^{\#}(P_t)$ (i.e., the quantity they are willing to buy at price $P_t$, which is the quantity that maximizes their net utility when the trading price is $P_t$) to the mediator.
5:   **if** $C_t \leq P_t$ **then** the retailer buys the quantity $U_t^{\#}(P_t)$ from the supplier at a total price of $P_t \cdot U_t^{\#}(P_t)$

---

The mediator's reward at every round $t$ is the so-called *gain from trade*, defined as the sum of the net gains of the supplier and retailer.[3] More precisely, when the supplier's cost is $c \in [0,1]$ and the buyer's utility function is $u \in$ USC, the gain from trade at a trading price $p \in [0,1]$ is defined as

$$g(p, c, u) := \big( \underbrace{(p - c) \cdot u^{\#}(p)}_{\text{supplier's net gain}} + \underbrace{u^{*}(p)}_{\text{retailer's net gain}} \big) \cdot \underbrace{\mathbb{I}\{c \leq p\}}_{\text{trade occurs}} .$$

---

[1]This is a strictly weaker condition than the commonly assumed Lipschitzness that is still sufficient for our theory.

[2]The reader might wonder why the mediator doesn't ask the two parties for their private cost and utility *before* proposing a trading price. The reasons for this choice have deep roots in mechanism design dating back to the Nobel-winning work of Myerson and Satterthwaite [1983]. We will address them in Section 2.

[3]Note that, from a regret-minimization perspective, this is equivalent to setting the *social welfare* as the mediator's objective.

The goal of the mediator is to minimize the *regret* defined, for any time horizon $T \in \mathbb{N}$, by

$$R_T := \sup_{p \in [0,1]} \mathbb{E}\left[ \sum_{t=1}^{T} g(p, C_t, U_t) - \sum_{t=1}^{T} g(P_t, C_t, U_t) \right] .$$

We develop our theory under four key assumptions (and will analyze the necessity of each of them).

**Assumption 1.1.** The model is stochastic: the sequence $(C, U)$, $(C_1, U_1)$, $(C_2, U_2), \ldots$ of supplier's costs and retailer's utility functions is i.i.d.

**Assumption 1.2.** The supplier's cost $C$ is independent of the retailer's utility function $U$.

**Assumption 1.3.** The cumulative distribution function (cdf) of the supplier's cost $C$ is $m$-Lipschitz, for some constant $m$.

**Assumption 1.4.** $\mathbb{E}\big[U^*(1)\big] = 0$; i.e., the expectation of the retailer's optimal utility at price 1 is 0.

In a real-world sense, Assumption 1.1 implies that the behaviors of suppliers/retailers follow the same general pattern over time without being influenced by past interactions. In well-known platforms, where the number of users is high, the platform rarely sees the same supplier/retailer within a short window, and the population of suppliers/retailers tends to be stable over time. This makes the i.i.d. assumption reasonable.

Assumption 1.2 simply means that the supplier's production cost *does not* directly influence the retailer's utility function. A first reason why this assumption is reasonable is related to the distinct economic drivers of suppliers and retailers. Supply-side costs typically depend on factors like input prices, technology, scale, and logistics, whereas demand-side utilities depend on consumer preferences, substitution patterns, and timing. These factors are often distinct from one another. For example, a fall in steel prices lowers bicycle-frame costs but does not necessarily change a commuter's willingness to pay for the bike. Another reason is that in many mediated markets, the item traded is fixed (e.g., a specific SKU, ride, or room-night). In this case, switching a supplier for another affects the production cost but not the retailer's valuation of that identical item. In other words, if it is always the same type of item that is being traded, it is realistic to assume that the random draw of the supplier would not influence that of the retailer. Hence, this assumption models the real-life fact that the utility earned by the retailer is only directly affected by the trading price.

Assumption 1.3 is a common, mild technical assumption that prevents highly concentrated distributions capable of generating pathological and unrealistic scenarios. In other words, it prevents large clusters of suppliers with costs concentrated in a narrow price range, which guarantees that small changes in the trading price lead to stable changes in the likelihood of the trade. As a counterexample, suppose that a large proportion of rideshare drivers in a population refuel at the same gas station in a region where fuel prices are relatively stable. In this case, many drivers have nearly identical costs, resulting in a sharp spike in the cdf of supplier costs around the shared fuel price, which is a violation of our assumption. As a result, a small change in the trading price could cause a large proportion of drivers to accept a trade, causing unstable or unpredictable outcomes. We would argue that these types of scenarios with highly concentrated distributions are not overly common in practice. Thus, the assumption of Lipschitzness is well-justified in many real-world scenarios.

Finally, Assumption 1.4 naturally states that if a good reaches the highest possible price of 1, then the retailer has no opportunity for profit and thus will not purchase any amount, resulting in zero utility. For example, a rideshare rider may open the Uber app to check whether the fare is below their personal threshold that they are willing to pay. At the maximum fare, it will exceed their threshold, making it likely the user will either walk or switch to a different service, such as Lyft. This behavior corresponds to a capped utility function, where utility vanishes beyond a certain price. One scenario in which this assumption is violated is when demand is inelastic: for instance, if a retailer must buy a certain quantity of the good no matter the cost. Following our previous example, the rider may be willing to purchase a ride at the maximum price if no alternative transportation options exist, e.g. if Uber is the only available service. That being said, such cases typically require a pure monopoly over a market or an emergency setting, which are relatively uncommon.

Hence, Assumptions 1.1–1.4 are well-motivated by real-world factors. We further investigate each assumption in Theorem 5.1, showing that they are not only justified but necessary for learnability.

## 1.3 Overview of our results

Our contributions are fourfold.

1. We prove that the mediated newsvendor mechanism is incentive compatible, individually rational, and budget balanced (Theorem 2.1).

2. We present an algorithm (Algorithm 1) for the repeated mediated newsvendor problem and prove that, under Assumptions 1.1–1.4, the algorithm achieves a $\widetilde{\mathcal{O}}(T^{2/3})$ regret (Theorem 3.3).

3. We prove the optimality of Algorithm 1 by establishing a matching lower bound of order $\Omega(T^{2/3})$, showing that its rate is unimprovable, up to logarithmic terms (Theorem 4.1).

4. We investigate the necessity of Assumptions 1.1–1.4, proving that lifting any one of them makes the problem unlearnable, even if the remaining three hold (Theorem 5.1).

## 1.4 Techniques and challenges

In this section, we outline the main challenges arising from the analysis of our setting, as well as the techniques we employed to overcome them.

**Uncountable action space.** First, the action space is uncountably infinite, a hurdle that usually prevents online learning techniques from working without further assumptions. One typical condition that enables online learning algorithms to work in such action spaces is that the expected reward function is Lipschitz, which opens the door to using discretization methods. However, in our setting, the expected reward function is not even *one-sided Lipschitz* (a weaker regularity condition that is still typically sufficient to guarantee learnability) without further assumptions. We overcome this obstacle by showing that, under the mild condition that the distribution of the supplier's cost is not too concentrated, we can in fact ensure that the expected reward function is one-sided Lipschitz, thereby unlocking discretization methods. Importantly, the validity of this mild assumption is not a limitation of our analysis, as we show that the problem becomes unlearnable when the cdf of the supplier's cost is not Lipschitz (Theorem 5.1).

**Severely limited feedback.** Even under the assumption that the cdf of the supplier's cost is Lipschitz, significant challenges remain. In particular, the mediator cannot implement standard bandit algorithms—despite the rich body of literature on this topic—because the available feedback is too limited to reconstruct standard bandit feedback. Indeed, whenever a trade occurs, the reward function equals the sum of the supplier's net gain $(P_t - C_t)U_t^{\#}(P_t)$ and the retailer's net gain $U_t^*(P_t)$. However, the mediator *only* learns a *threshold* about the supplier's cost $C_t$—specifically, that it is *below* the posted price $P_t$. Moreover, instead of observing the retailer's net gain $U_t^*(P_t)$, the mediator *only* receives the quantity $U_t^{\#}(P_t)$ that the retailer wishes to purchase at price $P_t$. Consequently, one must find a way to reconstruct the supplier's net utility from the threshold feedback and the retailer's net utility from the quantity purchased.

A first hint on how this feedback can be used comes from Step 1 of the proof sketch of Theorem 3.3, which indicates that when the supplier's cost is independent of the retailer's utility function, the expected reward function at any price $p \in [0, 1]$ is equal to $\mathbb{E}[U^{\#}(p)]\mathbb{E}[(p - C)^+] + \mathbb{E}[\mathbb{I}\{C \leq p\}]\mathbb{E}[U^*(p)]$. Thus, if one could reliably estimate the terms $\mathbb{E}[(p - C)^+]$ and $\mathbb{E}[U^*(p)]$, then the feedback available to the mediator after each interaction would in fact allow them to estimate the (expected) reward, turning the original problem into, essentially, a bandit problem. Under the assumption that the suppliers' costs and retailers' utilities form an i.i.d. process, we design a strategy where, first, we run an initial phase to construct optimistic estimates of the terms $\mathbb{E}[(p - C)^+]$ and $\mathbb{E}[U^*(p)]$, and then subsequently run a UCB-type bandit algorithm with the available feedback.

The last missing and crucial ingredient is to determine *how* to run the first phase to build these optimistic estimates for the terms that we do not have direct access to. To do this, we leverage the celebrated Envelope Theorem of Milgrom and Segal (see Appendix F) and show that, via an integration procedure, we can approximately and optimistically reconstruct the net utilities $\mathbb{E}[(p - C)^+]$ and $\mathbb{E}[U^*(p)]$ at any price $p$. This reconstruction uses the mediator's feedback by querying prices at points arranged on a uniform grid, which may be located far from $p$.

We conclude this section by remarking that the assumptions that led us to this strategy are not merely limitations of our analysis. Indeed, as shown in Theorem 5.1, carefully constructed impossibility results imply that relaxing any of them leads to unlearnability.

## 1.5 Related work

The newsvendor problem first appeared in Edgeworth [1888] and was later formalized by Arrow et al. [1951]—we refer the reader to Choi [2012] for a survey of the many variants of Arrow's model.

A game-theoretic formulation of the newsvendor problem with competing retailers was proposed by Parlar [1988]—see also Lippman and McCardle [1997], Mahajan and van Ryzin [2001], Netessine et al. [2006]. Wang and Gerchak [2003] use a Stackelberg game to model a situation where an assembler has to buy components from different suppliers. Lariviere and Porteus [2001] study a model where a supplier and a retailer interact through a price-only contract, and compare its efficiency with the efficiency of an integrated system. Adida and DeMiguel [2011] consider a competitive inventory model with several suppliers and several retailers, and prove equilibrium uniqueness under some symmetry conditions. We refer the reader to Cachon and Netessine [2006] for a survey of the literature on game-theoretic models in supply chain analysis and to Silbermayr [2020] for a more recent and specific survey on newsvendor games.

The mediated newsvendor problem is a strict generalization of the bilateral trade problem. The literature on bilateral trade is extremely rich and has experienced a steady growth since the fundamental work of Myerson and Satterthwaite [1983]. Classically, bilateral trade has been explored in the one-shot setting, mainly from a game-theoretic and approximation perspective [Colini-Baldeschi et al., 2016, 2017, Blumrosen and Mizrahi, 2016, Brustle et al., 2017, Colini-Baldeschi et al., 2020, Babaioff et al., 2020, Dütting et al., 2021, Deng et al., 2022, Kang et al., 2022, Archbold et al., 2023]. For a fairly complete overview on this literature, see, e.g., Cesa-Bianchi et al. [2023c]. On the other hand, a recent stream of literature has explored bilateral trade in a repeated setting through the lens of online learning. Given its greater relevance to our work, we focus on this literature.

In Cesa-Bianchi et al. [2021], Azar et al. [2022], Cesa-Bianchi et al. [2023c,b], Bernasconi et al. [2024], Cesa-Bianchi et al. [2024], Bacchiocchi et al. [2025], Gaucher et al. [2025], Bachoc et al. [2024], Bolić et al. [2024], Bachoc et al. [2025a,b], Cesari and Colomboni [2025], the authors examined the repeated bilateral trade problem where sellers and buyers trade non-divisible items. At each interaction with a new seller-buyer pair, the platform proposes a trading price, and the current item is traded if and only if the proposed price exceeds the seller's private valuation and is below the buyer's. When a trade occurs, the buyer pays the posted price to the seller, the seller transfers the item to the buyer, and the platform is rewarded with the gain from trade, i.e., the sum of the seller's and buyer's utility. In Cesa-Bianchi et al. [2021, 2023c], the authors investigated and obtained sharp regret bounds when sellers' and buyers' valuations for the non-divisible items being traded—represented by two random sequences of numbers $(S_t)_{t \in \mathbb{N}}, (B_t)_{t \in \mathbb{N}}$—form two i.i.d. sequences, while also showing that the adversarial case is unlearnable in general. Azar et al. [2022] managed to obtain learnability in the adversarial case by relaxing the notion of regret to that of 2-regret. When the platform can post two different prices to sellers and buyers, but is still not allowed to subsidize trades, Cesa-Bianchi et al. [2023b, 2024] achieved learnability using the usual notion of regret when the adversary belongs to the class of *smoothed* adversaries. Bernasconi et al. [2024] managed to achieve learnability in the adversarial case by allowing the platform to subsidize trade, as long as the subsidization comes from revenue obtained from previous seller and buyer interactions, Chen et al. [2025] improved on these results, and Lunghi et al. [2026] explored trade offs between budget violations and attainable regrets regimes. Bacchiocchi et al. [2025] investigated an asynchronous protocol for bilateral trade when sellers are queried only when a buyer is already secured. Gaucher et al. [2025] studied a contextual version of bilateral trade problem. Bachoc et al. [2024] proposed a variant of the gain from trade as reward to promote a fair division of profits among sellers and buyers. Babaioff et al. [2024], Lunghi et al. [2025] explored bilateral trade with multiple buyers. Bolić et al. [2024] investigated and characterized learnability in the i.i.d. setting when, at each time step, the two traders do not have predetermined seller and buyer roles, but can switch from one to the other depending on the proposed trading price. This last setting has been further investigated with contextual information in Bachoc et al. [2025a,b] and with the different objective of maximizing volume instead of gain from trade in Cesari and Colomboni [2025].

Finally, divisible items have also been recently proposed and studied in a different welfare maximization setting for dynamic pricing in Cesa-Bianchi et al. [2025].

## 2 Incentive compatibility, individual rationality, and budget balance

In this section, we show that the mediated newsvendor mechanism is incentive compatible, individually rational, and budget balanced—that is, the supplier and retailer gain nothing by misrepresenting their cost or utility function to the mediator, they never lose money by participating in a trade, and the mediator cannot subsidize or siphon welfare from trades.[4]

**Incentive compatibility.** We say that the mediated newsvendor mechanism is *incentive compatible* if, letting $c$ be the supplier's cost, $u$ the retailer's utility, and $p$ the proposed trading price, it is a *dominant strategy for the supplier* to accept a trade at price $p$ if and only if $c \leq p$, and it is a *dominant strategy for the retailer* to select the quantity $u^{\#}(p)$.

**Individual rationality.** We say that the mediated newsvendor mechanism is *individually rational* if the net utilities of the supplier and the retailer are non-negative. Here, given a production cost $c$, an order quantity $q$, and a trading price $p$, the *net utility of the supplier* is defined as $(qp - qc)\mathbb{I}\{c \leq p\}$, and the *net utility of the retailer* is defined as $u(q) - pq$.

**Budget balance.** We say that the mediated newsvendor mechanism is *budget balanced* if, whenever a payment of $x$ needs to be transferred from the retailer to the supplier, the retailer is asked for exactly $x$, and the sum $x$ is transferred in its entirety to the supplier.

**Theorem 2.1.** *The mediated newsvendor mechanism is incentive compatible, individually rational, and budget balanced.*

*Proof sketch (full proof in Appendix B).* We separately prove the three claims in the statement of Theorem 2.1. To show incentive compatibility, we prove that any attempt by either the retailer or the supplier to misreport their private information—whether by over- or under-reporting it—leads to a decrease in their respective utilities. Next, individual rationality follows directly from the fact that suppliers will never accept a trade when their costs exceed the offered price, and that retailers are assumed to optimize their net utility, ensuring it is always non-negative (since they always have the option to abstain from purchasing, resulting in a utility of 0). Finally, the budget-balanced property can be derived from its definition within the context of this setting. □

We remark that incentive compatibility, individual rationality, and budget balance are three of the four classic *desiderata* in bilateral trade mechanisms like the newsvendor problem. The fourth one is *market efficiency*, that is, the ability to make the supplier and retailer trade every time they have the opportunity to do so—i.e., whenever there exists a $p$ such that $c \leq p$ and $u^{\#}(p) > 0$. The Nobel-winning work on bilateral trade of Myerson and Satterthwaite [1983] shows that, in general, the above four properties cannot hold simultaneously. Therefore, Theorem 2.1 yields a *maximal* set of properties that any bilateral trade mechanism can satisfy and is thus unimprovable.

## 3 A no-regret algorithm for RMNP

In this section, we present our Algorithm 1 and provide an upper bound on its regret.

To lighten the notation, we begin by introducing the following confidence radii that we repeatedly use. For each $K \in \mathbb{N}$, $\delta \in (0, 1)$, and $T, n \in \mathbb{N}$, we define

$$\zeta_{K,\delta} := \frac{1 + \sqrt{2\ln(8K/\delta)}}{K} \qquad \text{and} \qquad \xi_{T,K,\delta}(n) := \sqrt{\frac{2\ln(8KT/\delta)}{n}} \ .$$

---

[4]Note that he result does not follow directly from Assumption 1.1, since even in an i.i.d. setting, participants can benefit from misreporting their valuations. E.g., in first-price auctions, bidders have an incentive to under-report despite i.i.d. draws and one-shot participation. Thus, it is essential to analyze the mechanism, not just the generation of valuations, to assess incentive compatibility, individual rationality, and budget balance.

We are now ready to present our algorithm. Algorithm 1 takes the time horizon $T$, a discretization parameter $K \in \mathbb{N}$ and a confidence parameter $\delta \in (0,1)$ as input. Then, it builds a grid of $(K+1)$-equispaced points $0 = p_0 < p_1 < \cdots < p_K = 1$ and spends the first $K^2$ rounds by posting the $K$ prices $p_0, p_1, \ldots, p_{K-1}$ in a round-robin fashion. Afterwards, for any $j \in \{0, 1, \ldots, K-1\}$, it uses this information to build upper confidence bounds $F_j + \zeta_{K,\delta}$ (resp. $G_j + \zeta_{K,\delta}$) for $\mathbb{E}[U^*(p_j)]$ (resp. $\mathbb{E}[(p_j - C)^+]$). These two terms will be useful to build estimates of the two multiplicative terms we will use to reconstruct the expected reward $p \mapsto \mathbb{E}[g(p, C, U)]$ when evaluated at the points $p_0, p_1, \ldots, p_{K-1}$. After those $K^2$ rounds, the algorithm follows an *upper confidence bound* strategy on the reward function, leveraging the previous estimates, and the bandit feedback it collects along the way.

---

**Algorithm 1**

---

**input:** Time horizon $T \in \mathbb{N}$, discretization parameter $K \in \mathbb{N}$, and confidence parameter $\delta \in (0,1)$
**init:** Set $\forall j \in \{0, \ldots, K\}$, $p_j := \frac{j}{K}$, $A_j(0) := B_j(0) := N_j(0) := 0$
  1: **for** each time $t = 1, 2, \ldots$ **do**
  2:    **if** $t \leq K^2$ **then** select $I_t := t - 1 \pmod K$
  3:    **if** $t > K^2$ **then**
  4:        Update $\forall j \in \{0, \ldots, K-1\}$, $\hat{g}_{t-1}(j) := \left( \frac{A_j\left(N_j(t-1)\right)}{N_j(t-1)} + \xi_{T,K,\delta}\left(N_j(t-1)\right) \right) (G_j + \zeta_{K,\delta})$

$$\qquad\qquad\qquad + \left( \frac{B_j\left(N_j(t-1)\right)}{N_j(t-1)} + \xi_{T,K,\delta}\left(N_j(t-1)\right) \right) (F_j + \zeta_{K,\delta})$$

          where, $\forall j \in [K]$, $F_j := \sum_{i=j-1}^{K-1} \frac{A_i\left(N_i(K^2)\right)}{K^2}$, and $\forall j \in [K-1]$, $G_j := \sum_{i=0}^{j-1} \frac{B_i\left(N_i(K^2)\right)}{K^2}$
  5:        Select $I_t \in \operatorname{argmax}_{j \in \{0, \ldots, K-1\}} \hat{g}_{t-1}(j)$
  6:    Post price $P_t := p_{I_t}$ and receive feedback $\mathbb{I}\{C_t \leq P_t\}$ and $U_t^{\#}(P_t)$
  7:    Update $\forall j \in \{0, \ldots, K-1\}$, $N_j(t) := N_j(t-1) + \mathbb{I}\{j = I_t\}$,
        $A_{I_t}\left(N_{I_t}(t)\right) := A_{I_t}\left(N_{I_t}(t-1)\right) + U_t^{\#}(P_t)$, $B_{I_t}\left(N_{I_t}(t)\right) := B_{I_t}\left(N_{I_t}(t-1)\right) + \mathbb{I}\{C_t \leq P_t\}$

---

To provide regret guarantees for Algorithm 1, we begin by proving a lemma that allows us to relate the retailer's optimal net gain $u^*$ (that appears in the gain for trade) with the retailer's desired quantity function $u^{\#}$ (that appears in the feedback the learner receives), for any (gross) utility function $u \in \mathrm{USC}$.

**Lemma 3.1.** *If $u \in \mathrm{USC}$, then, for any selection of $u^{\#}$ and any $0 \leq p_1 < p_2 \leq 1$, it holds that*

$$u^*(p_1) - u^*(p_2) = \int_{p_1}^{p_2} u^{\#}(\lambda)\, \mathrm{d}\lambda \,.$$

*Proof sketch (full proof in Appendix C).* The key idea of the proof is leveraging Milgrom and Segal's Envelope Theorem, which establishes an integral representation of some transformation of a function in terms of another transformation of the same function. A careful derivation applied to the transformations $u^*$ and $u^{\#}$ of $u$ leads to the result. □

Before stating and proving our regret upper bound for Algorithm 1 (Theorem 3.3), we require a second technical lemma, whose economic interpretation is that the retailer's net gain decreases as the price increases, and so does the quantity that the retailer decides to buy.

**Lemma 3.2.** *For any $u \in \mathrm{USC}$, the functions $u^*$ and $u^{\#}$ are monotonically non-increasing.*

*Proof sketch (full proof in Appendix C).* The monotonicity of $u^*$ is a direct consequence of the previous Lemma 3.1. The same property can be proven for $u^{\#}$ through a derivation involving a lower bounding, for any $0 \leq p_1 \leq p_2 \leq 1$, on the product $(p_2 - p_1)\left(u^{\#}(p_1) - u^{\#}(p_2)\right)$, by exploiting some elementary properties of $u^*$ and $u^{\#}$. □

**Theorem 3.3.** *Assume that Assumptions 1.1–1.4 hold. If we run Algorithm 1 with time horizon $T$, discretization parameter $K := \lceil T^{1/3} \rceil$ and confidence parameter $\delta := \frac{1}{T}$, then*

$$R_T = \widetilde{O}\left(T^{2/3}\right) \,.$$

*Proof sketch (full proof in Appendix C).* The proof rests on four key ideas: 1. Although the learner does not observe the *realized* reward gained when posting a price, the *expected* reward associated with that price can be expressed as a function of terms that can be estimated either directly or indirectly, 2. An initial phase can be used to estimate the terms that cannot be directly observed, 3. The discretization error can be carefully controlled, and 4. The regret suffered during the second (and main) phase can be managed by leveraging the feedback relative to the terms that can be directly observed.

*Step 1. Understanding the expected reward.* The keystone of this analysis is rewriting the expected reward at a price $p$ as $\mathbb{E}\big[U^{\#}(p)\big]\,\mathbb{E}\big[(p-C)_+\big] + \mathbb{E}\big[\mathbb{I}\{C \leq p\}\big]\,\mathbb{E}\big[U^*(p)\big]$. Recall that the feedback received by posting a price $p$ at time $t$ is $U_t^{\#}(p)$ and $\mathbb{I}\{C_t \leq p\}$. Thus, the two terms $\mathbb{E}\big[U^{\#}(p)\big]$ and $\mathbb{E}\big[\mathbb{I}\{C \leq p\}\big]$ can be directly estimated using the observed feedback, whereas the terms $\mathbb{E}\big[(p-C)_+\big]$ and $\mathbb{E}\big[U^*(p)\big]$ cannot.

*Step 2. Estimating the "hidden" terms.* Algorithm 1 spends the first $K^2$ rounds cycling over the $K$ grid points $p_0, \ldots p_{K-1}$ and gathering feedback samples. These samples can be used in a non-trivial way to build high-probability, optimistic estimates of $\mathbb{E}\big[(p-C)_+\big]$ and $\mathbb{E}\big[U^*(p)\big]$ for every grid point $p$.

*Step 3. Controlling the discretization error.* The discretization error can be controlled by combining our two Lemmas 3.1 and 3.2 with the Lipschitzness assumption on the supplier's cdf, obtaining, for any $j \in \{0, \ldots, K\}$ and any $p \in [p_j, p_{j+1}]$, that $\big|f(p) - f(p_j)\big| = O\big(\frac{1}{K}\big)$.

*Step 4. Estimating the "observable" terms.* In the second (and main) phase, Algorithm 1 exploits the optimistic estimates of the "hidden" terms built in the initial phase, putting them together with natural UCB estimates of the "observable" terms that are built sequentially. This combination provides high-probability guarantees for the estimator $\hat{g}_{t-1}$ maintained on Line 4 of the algorithm, uniformly across all points in the grid. More precisely, with probability at least $1 - \delta$, it holds that, for all

$$j \in \{0, \ldots, K\}, \ f(p_j) \leq \hat{g}_{t-1}(j) \leq f(p_j) + 16\sqrt{2\ln\big(\tfrac{8KT}{\delta}\big)}\big/\min\big(\sqrt{N_j(t-1)}, K\big).$$

*Putting everything together.* Combining the four steps above, one can prove that the regret can be upper bounded as follows

$$R_T = O\left(T\delta + K^2 + \frac{T}{K} + \sqrt{\log\left(\frac{KT}{\delta}\right)}\left(\frac{T}{K} + \sqrt{KT}\right)\right)$$

which, plugging in the values of $K = \lceil T^{1/3} \rceil$ and $\delta = \frac{1}{T}$, yields the desired result. $\qquad\square$

## 4 Optimality of Algorithm 1

In this section, we show that, up to logarithmic terms, the performance of Algorithm 1 cannot be improved.

**Theorem 4.1.** *Assume that Assumptions 1.1–1.4 hold. Then, the worst-case regret of any algorithm satisfies*

$$R_T = \Omega\big(T^{2/3}\big)\,.$$

*Proof sketch (full proof in Appendix D).* The key insight to obtain this result is that our setting is a proper generalization of repeated bilateral trade (we prove this formally in Appendix A), a setting where the retailer's decision is restricted to quantities $Q_t \in \{0, 1\}$, where $Q_t := \mathbb{I}\{P_t \leq B_t\}$ and $B_t$ is the retailer's *private valuation* for the good on sale. In particular, we show that repeated bilateral trade can be seen as a particular instance of the RMNP when the retailer's utility function $U_t$ is a linear function from $[0, 1]$ to $[0, 1]$ with slope $B_t$. Once this reduction has been established, we can leverage tools from bilateral trade to obtain the desired lower bound. $\qquad\square$

As we mentioned in the proof sketch of Theorem 4.1, the RMNP is a proper generalization of the better-understood repeated bilateral trade problem. Consequently, our algorithm and upper bound in Section 3 apply to the bilateral trade problem too, yielding near-optimal guarantees in this more restrictive case as well. We remark that it is typically not true that an algorithm with near-optimal guarantees in an online learning problem is also near-optimal in all special cases of the general

problem because, in general, algorithms tailored to special cases can exploit this knowledge to obtain better performance. Theorems 3.3 and 4.1 show that, instead, our Algorithm 1 automatically adapts to bilateral trade, retaining its near-optimality.

# 5  Necessity of assumptions

The reader may wonder whether Assumptions 1.1–1.4 are necessary for learnability. In this section, we present a strong negative result showing that the learning problem becomes unlearnable if even one of these assumptions is violated, regardless of whether the other three still hold.

**Theorem 5.1.** *Assume that at least one of Assumptions 1.1–1.4 does not hold. Then, even if the remaining three hold, the worst-case regret of any algorithm satisfies*

$$R_T = \Omega(T).$$

*Proof sketch (full proof in Appendix E).* The four points are proved separately. For each one, we build a family of "hard" instances, each satisfying exactly three of the four Assumptions 1.1–1.4. We then show that every algorithm suffers linear regret in at least one instance of this family. □

# 6  Experiments

In this section, we empirically validate our theoretical results by simulating a variety of supplier cost distributions and retailer utility functions. Our goal is to illustrate the practical effectiveness of our algorithm and the necessity of Assumptions 1.1–1.4.

To validate our theoretical upper bound, we evaluate the regret of Algorithm 1 across several supplier cost distributions designed to represent realistic mediated market conditions. Firstly, we use the Uniform(0,1) distribution as a neutral baseline for our experiments. Then, to model markets where low-cost suppliers are prevalent but occasional high-cost suppliers exist, we consider two right-skewed distributions: a Beta($\alpha = 2$, $\beta = 5$) and a Log-Normal($\mu = -0.5$, $\sigma = 1$) truncated to $[0, 1]$, with the latter assigning more probability to high-cost suppliers. Finally, to represent markets with two distinct tiers of suppliers, we include a bimodal mixture $0.75 \cdot \text{Beta}(2, 5) + 0.25 \cdot \text{Beta}(5, 2)$, which creates a majority group of low-cost suppliers and a smaller, higher-cost group.

We also consider two realistic and prevalent families of retailer utility functions. Firstly, we model satiable demand through the capped-linear utility function $U_{a,\bar{q}}(q) = \min\{aq, a\bar{q}\}$, where the retailer's utility grows at a constant marginal rate $a \in [0, 1]$ until the quantity purchased reaches the saturation threshold $\bar{q} \in [0, 1]$, after which purchasing additional goods generates no further benefit. For our experiments, we draw $a$ from a Beta(5, 2) distribution and $\bar{q}$ from a Beta(2,2) distribution. In this scenario, retailers generally place a high value on initial purchases until inventory reaches a moderate level, at which point acquiring more goods yields no extra value to the retailer.

Secondly, we model diminishing marginal returns using the exponential utility function $U_\lambda(q) = \left(1 - e^{-\lambda q}\right)/\lambda$, where $\lambda > 0$ determines the overall valuation level. In this case, the marginal gain from purchasing additional goods decreases progressively as the quantity increases. For our experiments, we draw $\lambda$ from a Log-Normal$(0, 0.5)$ distribution, resulting in retailers whose initial marginal valuations are generally moderate and whose marginal benefits decline smoothly as the quantity purchased increases.

Note that all supplier and retailer distributions described above satisfy Assumptions 1.1–1.4, meaning that we expect that $R_T = \widetilde{O}(T^{2/3})$ by Theorem 3.3.

As expected, Algorithm 1 behaves according to the theoretical guarantees we established across all the combinations of supplier and retailer distributions described above (see Figure 1).[5] While the regret remains similar across seller distributions for a given utility, the capped-linear utility appears to have a moderately lower growth rate in regret than the exponential utility.

We utilize the distributions defined in the proof of Theorem 5.1 in Appendix E to illustrate the linear lower bounds empirically. In the proof sections corresponding to Assumptions 1.2 and 1.4, we relax

---

[5]For reference, we ran our experiments on a MacBook Pro with an M1 Pro chip (10-core CPU, 16-core GPU) and 16 GB of RAM. With the given configuration, each plot takes at most around 3 minutes to generate.

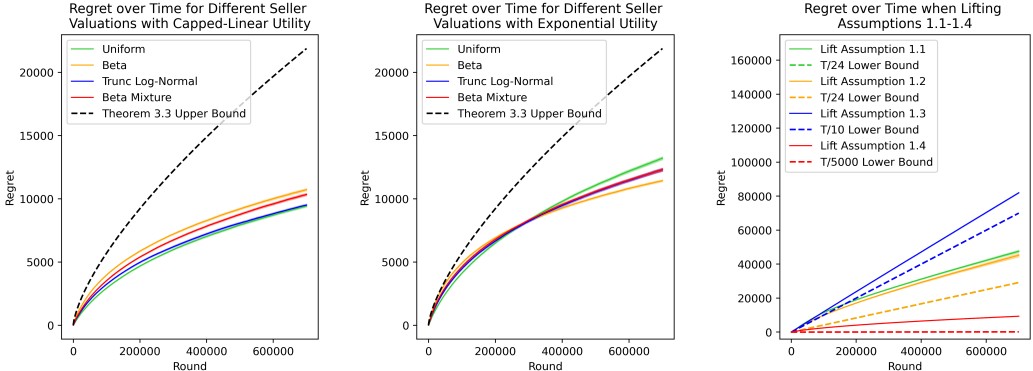

Figure 1: The $x$-axis represents the time horizon $T$, and the $y$-axis represents the regret $R_T$. Each curve shows the mean regret over 30 random trials, with the shaded bands around the curves representing the 95% confidence intervals on the regret computed under the normality assumption. The left and center plots illustrate the algorithm's performance across various supplier cost distributions under capped-linear and exponential retailer utilities, respectively. They include the theoretical upper bound from Theorem 3.3 (scaled down for visualization). The right plot shows the algorithm's performance when each of the assumptions is removed, using the distributions from the proof of Theorem 5.1 in Appendix E, along with their lower bounds.

both assumptions in turn and construct two environments such that any algorithm will suffer linear regret under at least one of them. We sample from the first environment in each pair to generate the empirical results. For Assumption 1.1, the proof constructs two families of adversarial environments, adapted from those in Assumption 1.2, and shows that each family's worst-case regret is at least as large as the regret in the corresponding Assumption 1.2 environment. Hence, we select an adversary from the first family to generate the empirical regret of Algorithm 1 with respect to a sequence of supplier costs and retailer utility functions that is not i.i.d. For Assumption 1.3, the proof defines an infinite family of environments with non-Lipschitz supplier cost cdfs, showing that every learning algorithm suffers linear regret under at least one. Thus, we select a fixed environment from this family and display its regret in Figure 1. As expected, in each of the four configurations where one of the Assumptions 1.1–1.4 is lifted, the regret of Algorithm 1 grows linearly (see Figure 1).

# 7  Conclusions, limitations, and future directions

Our paper explores a repeated version of the mediated newsvendor problem. After proving that the mediated newsvendor mechanism is incentive compatible, individually rational, and budget balanced, we provide a comprehensive analysis of the repeated mediated newsvendor problem consisting of matching upper and lower regret bounds of order $T^{2/3}$ under four key assumptions, along with impossibility results if any of the assumptions are removed.

Our research encourages further exploration of this setting. A minor limitation of our work is that our regret guarantees in Theorem 3.3 are tight up to logarithmic factors, hence further research is needed to determine the precise upper or lower bound. Additionally, one could investigate a *fair* variant of this setting in the spirit of Bachoc et al. [2024], where the instantaneous objective of the learner is the minimum between the supplier's and retailer's utilities rather than their sum. A further avenue could be to explore a contextual setting where a context vector is available to the learner at each round, to capture shared shocks (e.g., seasonal effects) that influence both suppliers and retailers. Another interesting extension could be to examine a weakly-budget balanced version of this setting where the learner can post two potentially distinct prices to the retailer and the supplier. A final intriguing line of research would be to consider a bargaining mechanism for the supplier where, instead of simply accepting/rejecting the proposed trading price, they are allowed to counter. We leave all these interesting directions to future research.

## Acknowledgments and Disclosure of Funding

NB, CP, and TC acknowledge the support of NSERC through USRA (Undergraduate Student Research Awards).

TC also gratefully acknowledges the support of the University of Ottawa through grant GR002837 (Start-Up Funds) and that of the Natural Sciences and Engineering Research Council of Canada (NSERC) through grants RGPIN-2023-03688 (Discovery Grants Program) and DGECR2023-00208 (Discovery Grants Program, DGECR - Discovery Launch Supplement).

RC is partially supported by the MUR PRIN grant 2022EKNE5K (Learning in Markets and Society), the FAIR (Future Artificial Intelligence Research) project, funded by the NextGenerationEU program within the PNRR-PE-AI scheme, the EU Horizon CL4-2022-HUMAN-02 research and innovation action under grant agreement 101120237, project ELIAS (European Lighthouse of AI for Sustainability).

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

# A    RMNP is a generalization of bilateral trade (For Section 1)

**Theorem A.1.** *The repeated mediated newsvendor problem is a proper generalization of the repeated bilateral trade problem.*

*Proof.* To prove this claim, it suffices to show that any instance of the bilateral trade problem can be mapped to an instance of the RMNP such that the feedback and reward under both environments is identical.

To do this, consider an arbitrary instance of the bilateral trade problem, denoted by $(S_t, B_t)_{t \in \mathbb{N}}$. We define a corresponding instance $(C_t, U_t)_{t \in \mathbb{N}}$ of the RMNP as follows: $\forall t \in \mathbb{N}$, let $C_t := S_t$ and let $U_t \in \text{USC}$ be defined, $\forall q \in [0, 1]$, by $U_t(q) := B_t \cdot q$.

First, we observe that the associated RMNP problem has the same action space $[0, 1]$ as the bilateral trade problem.

Next, we show that, for any posted price $p \in [0, 1]$, the learner receives the same feedback under the instance $(C_t, U_t)_{t \in \mathbb{N}}$ as under $(S_t, B_t)_{t \in \mathbb{N}}$.

Suppose that the learner posts a trading price $p \in [0, 1]$ at time $t \in \mathbb{N}$. Then, in the case of bilateral trade, the learner observes $\mathbb{I}\{S_t \leq p\}$ from the seller and $\mathbb{I}\{p \leq B_t\}$ from the buyer. In the corresponding RMNP instance, the learner observes $\mathbb{I}\{C_t \leq p\} = \mathbb{I}\{S_t \leq p\}$ from the supplier and

$$U_t^\#(p) \in \underset{q \in [0,1]}{\text{argmax}}\, (U_t(q) - p \cdot q) = \underset{q \in [0,1]}{\text{argmax}}\, (B_t \cdot q - p \cdot q) = \begin{cases} \{1\} & \text{if } p < B_t \\ \{0\} & \text{if } p > B_t \\ [0, 1] & \text{if } p = B_t \end{cases}$$

from the retailer. If we assume that in the case of a tie (i.e., $B_t = p$), the retailer selects the maximum quantity possible, namely 1, then the observed feedback is precisely

$$U_t^\#(p) = \max\left( \underset{q \in [0,1]}{\text{argmax}}\, (U_t(q) - p \cdot q) \right) = \begin{cases} 1 & \text{if } p \leq B_t \\ 0 & \text{if } p > B_t \end{cases} = \mathbb{I}\{p \leq B_t\}.$$

Thus, the feedback received under $(C_t, U_t)_{t \in \mathbb{N}}$ is identical to that received under $(S_t, B_t)_{t \in \mathbb{N}}$.

Finally, we show that the reward function under $(C_t, U_t)_{t \in \mathbb{N}}$ is the same as the reward function under $(S_t, B_t)_{t \in \mathbb{N}}$.

Fix any $p \in [0, 1]$ and any $t \in \mathbb{N}$. In the case of the bilateral trade problem, the reward function under $(S_t, B_t)_{t \in \mathbb{N}}$ is defined as

$$\text{GFT}(p, S_t, B_t) = (B_t - S_t)\, \mathbb{I}\{S_t \leq p \leq B_t\}.$$

In the case of the RMNP, the reward function under $(C_t, U_t)_{t \in \mathbb{N}}$ is defined as

$$g(p, C_t, U_t) = \left( (p - C_t) \cdot U_t^\#(p) + U_t^*(p) \right) \cdot \mathbb{I}\{C_t \leq p\}.$$

Now, for every $p \in [0, 1]$, observe that

$$U_t^*(p) = U_t\left( U_t^\#(p) \right) - p \cdot U_t^\#(p) = B_t \cdot U_t^\#(p) - p \cdot U_t^\#(p) = (B_t - p)\, \mathbb{I}\{p \leq B_t\}.$$

Hence,

$$\begin{aligned}
g(p, C_t, U_t) &= \left( (p - C_t) \cdot U_t^\#(p) + U_t^*(p) \right) \cdot \mathbb{I}\{C_t \leq p\} \\
&= \left( (p - S_t)\, \mathbb{I}\{p \leq B_t\} + (B_t - p)\, \mathbb{I}\{p \leq B_t\} \right) \cdot \mathbb{I}\{S_t \leq p\} \\
&= (B_t - S_t)\, \mathbb{I}\{p \leq B_t\} \mathbb{I}\{S_t \leq p\} \\
&= (B_t - S_t)\, \mathbb{I}\{S_t \leq p \leq B_t\} \\
&= \text{GFT}(p, S_t, B_t).
\end{aligned}$$

Therefore, the reward function under $(C_t, U_t)_{t \in \mathbb{N}}$ is identical to that under $(S_t, B_t)_{t \in \mathbb{N}}$. Since $(S_t, B_t)_{t \in \mathbb{N}}$ is an arbitrary instance of the bilateral trade problem, it follows that the bilateral trade problem is a special case of the RMNP. Equivalently, the RMNP is a proper generalization of the bilateral trade problem. $\square$

# B  Missing details for Section 2

**Theorem B.1** (Theorem 2.1, restated). *The mediated newsvendor mechanism is incentive compatible, individually rational, and budget balanced.*

*Proof.* Let $c$ be the supplier's cost, $u$ the retailer's utility, $p$ the proposed trading price, $\widetilde{b}$ the decision disclosed by the supplier at price $p$ (1 if accepted, 0 if refused), and $\widetilde{q}$ the quantity disclosed by the retailer at price $p$.

**Incentive Compatibility.**  We begin by proving that the mediated newsvendor mechanism is incentive compatible.

- **Supplier's incentive.** Consider the supplier's possible decisions:

  If $\widetilde{b} = 0$, the supplier gains nothing. However, if $c < p$, the supplier could achieve positive gains by accepting the trade, i.e., by reporting $\widetilde{b} = 1$. Hence, refusing the trade when $c < p$ can only decrease the supplier's net utility.

  If $\widetilde{b} = 1$, the supplier's utility is $p - c$, which is negative when $p < c$. In contrast, by refusing the trade (i.e., reporting $\widetilde{b} = 0$), the utility is zero. Thus, accepting a trade when $p < c$ can only decrease the supplier's net utility.

  Finally, if $p = c$, the supplier is indifferent between accepting and refusing the trade, as both yield zero utility.

  Therefore, reporting truthfully, i.e., $\widetilde{b} = \mathbb{I}\{c \leq p\}$, is a dominant strategy for the supplier.

- **Retailer's incentive.** Next, we show that no deviation from $\widetilde{q} = u^{\#}(p)$ improves the retailer's utility. By definition:

$$u^{\#}(p) \in \operatorname*{argmax}_{q \in [0,1]} \big(u(q) - p \cdot q\big).$$

  Thus, for any deviation $\widetilde{q} \neq u^{\#}(p)$, we have:

$$u(\widetilde{q}) - p \cdot \widetilde{q} \leq u\big(u^{\#}(p)\big) - p \cdot u^{\#}(p).$$

  Additionally, the retailer's choice of quantity $\widetilde{q}$ does not affect the supplier's acceptance decision, which depends solely on the mediator's chosen price $p$. Therefore, reporting any $\widetilde{q}$ that deviates from the truthful choice $u^{\#}(p)$ can only decrease the retailer's utility.

**Individual Rationality.**  Next, we show that the mechanism is individually rational for both parties.

- **Supplier's rationality.** The supplier's net utility when a trade does not occur ($p < c$) is zero, while when a trade does occur ($c \leq p$), their net utility is:

$$(qp - qc)\mathbb{I}\{c \leq p\} = q(p - c)\mathbb{I}\{p - c \geq 0\} \geq 0.$$

  Thus, the supplier never incurs negative utility.

- **Retailer's rationality.** Since the retailer selects quantity:

$$u^{\#}(p) \in \operatorname*{argmax}_{q \in [0,1]} \big(u(q) - p \cdot q\big),$$

  it follows that:

$$u(u^{\#}(p)) - p \cdot u^{\#}(p) \geq u(0) - p \cdot 0 = u(0) \geq 0.$$

  Thus, the retailer also never incurs negative utility.

**Budget Balance.**  Lastly, we observe that the mechanism is budget balanced: the total amount the retailer pays is exactly the amount transferred to the supplier. Specifically, the supplier receives precisely the retailer's payment $p \cdot u^{\#}(p)$. $\qquad\square$

## C Missing details for Section 3

**Lemma C.1** (Lemma 3.1, Restated). *If $u \in$ USC, then, for any selection of $u^{\#}$ and any $0 \le p_1 < p_2 \le 1$ it holds that*

$$u^*(p_1) - u^*(p_2) = \int_{p_1}^{p_2} u^{\#}(\lambda) \, d\lambda \, .$$

*Proof.* This is a corollary of the celebrated Envelope Theorem [Milgrom and Segal, 2002]. For completeness, we report a version of this result as Theorem F.1 in Appendix F. Using the notation from Theorem F.1, define the function $f \colon [0,1]^2 \to \mathbb{R}$ by $(q,p) \mapsto u(q) - pq$. We observe that, for every $p \in [0,1]$, the function $q \mapsto f(q,p)$ is upper semi-continuous (being the sum of two upper semi-continuous functions). Furthermore, the function $(q,p) \mapsto \partial_p f(q,p) = -q$ is continuous. Hence, for any selection $x^\circ \colon [0,1] \to [0,1]$ satisfying, for all $p \in [0,1]$,

$$x^\circ(p) \in \underset{q \in [0,1]}{\operatorname{argmax}} f(q,p),$$

the function $V \colon [0,1] \to [0,1], p \mapsto \max_{q \in [0,1]} f(q,p)$ satisfies, for all $p \in [0,1]$,

$$V(p) = V(0) + \int_0^p \partial_p f\big(x^\circ(\lambda), \lambda\big) \, d\lambda \, .$$

The conclusion then follows by observing the following: for every $\lambda \in [0,1]$, we have $\partial_p f\big(x^\circ(\lambda), \lambda\big) = -x^\circ(\lambda)$; for every $p \in [0,1]$, we have $u^*(p) = V(p)$; and for every $p \in [0,1]$, $u^{\#}$ is a selection satisfying $u^{\#}(p) \in \operatorname{argmax}_{q \in [0,1]} f(q,p)$ . $\qquad\square$

**Lemma C.2** (Lemma 3.2, Restated). *For any $u \in$ USC, the functions $u^*$ and $u^{\#}$ are monotonically non-increasing.*

*Proof.* From Lemma 3.1, for any $0 \le p_1 < p_2 \le 1$, we have that $u^*(p_1) - u^*(p_2) = \int_{p_1}^{p_2} u^{\#}(\lambda) \, d\lambda \ge 0$, where the inequality follows from the non-negativity of $u^{\#}$. This directly implies that $u^*$ is monotonically non-increasing.

Now, to show that $u^{\#}$ is monotonically non-increasing, consider any $0 \le p_1 < p_2 \le 1$ and let $q_i := u^{\#}(p_i)$ for $i = 1, 2$. Then,

$$
\begin{aligned}
(p_2 - p_1)(q_1 - q_2) &= (u(q_2) - p_2 q_2) - (u(q_1) - p_2 q_1) + (u(q_1) - p_1 q_1) - (u(q_2) - p_1 q_2) \\
&= \underbrace{\max_{q \in [0,1]} (u(q) - p_2 q) - (u(q_1) - p_2 q_1)}_{\ge 0} + \underbrace{\max_{q \in [0,1]} (u(q) - p_1 q) - (u(q_1) - p_1 q_1)}_{\ge 0} \\
&\ge 0,
\end{aligned}
$$

and the conclusion follows upon dividing by $p_2 - p_1 > 0$. $\qquad\square$

**Theorem C.3** (Theorem 3.3, Restated). *Assume that Assumptions 1.1–1.4 hold. If we run Algorithm 1 with time horizon $T$, discretization parameter $K := \lceil T^{1/3} \rceil$ and confidence parameter $\delta := \frac{1}{T}$, then*

$$R_T = \widetilde{O}\big(T^{2/3}\big) \, .$$

*Proof.* Define, for any $p \in [0,1]$, $f(p) := \mathbb{E}\big[g(p, C, U)\big]$. Recalling Assumption 1.1, for any time horizon $T \in \mathbb{N}$, we define the pseudo-regret $\mathcal{R}_T$ suffered by a sequence of prices $P_1, \ldots, P_T$ up to the time horizon $T \in \mathbb{N}$ as

$$\mathcal{R}_T := \sum_{t=1}^{T} \left( \sup_{p \in [0,1]} f(p) - f(P_t) \right) \, .$$

Then, after a straightforward application of the Freezing Lemma F.2, we obtain that $R_T = \mathbb{E}\left[\mathcal{R}_T\right]$. This observation suggests studying the pseudo-regret to prove the regret bounds.

To prove the theorem, without any loss of generality, we may assume $T$ is sufficiently large. Specifically, we notice that the crude bound $T \ge 5 \cdot 10^7$ suffices for the following calculations. We leave the optimization of the involved constants to the interested reader.

Let $K \in \mathbb{N}$ with $2 \leq K$ and $K^2 < T$. We may assume that $(C_t, U_t)_{t \in \mathbb{N}}$ is generated in the following way. Consider two i.i.d. families $(C_{t,k})_{t \in \mathbb{N}, k \in \{0,1,\ldots,K-1\}}$ and $(U_{t,j})_{t \in \mathbb{N}, j \in \{0,1,\ldots,K-1\}}$ that are independent of each other. For every $t \in \mathbb{N}$ and every $j \in \{0, 1, \ldots, K-1\}$, the distribution of the pair $(C_{t,j}, U_{t,j})$ is identical to the distribution of $(C, U)$. At time $t = 1$, we set

$$(C_1, U_1) := (C_{N_{I_1}(0), I_1}, U_{N_{I_1}(0), I_1}).$$

Then, inductively, at each time $t + 1 \in \mathbb{N}$, when the algorithm selects the index $I_{t+1} \in \{0, 1, \ldots, K - 1\}$, we set

$$(C_{t+1}, U_{t+1}) := (C_{N_{I_t}(t)+1, I_{t+1}}, U_{N_{I_t}(t)+1, I_{t+1}}).$$

By induction, the sequence $(C_t, U_t)_{t \in \mathbb{N}}$ is well-defined. Moreover, this sequence is i.i.d., and the distribution of $(C_t, U_t)$ coincides with that of $(C, U)$ for every $t \in \mathbb{N}$.

Now, notice that by applying Assumption 1.2, for every $p \in [0, 1]$, we obtain that

$$f(p) = \mathbb{E}\big[U^\#(p)\big]\mathbb{E}\big[(p - C)^+\big] + \mathbb{E}\big[\mathbb{I}\{C \leq p\}\big]\mathbb{E}\big[U^*(p)\big] .$$

Then, for any $j \in \{0, 1, \ldots, K - 1\}$, if $p \in [p_j, p_{j+1}]$, leveraging Lemma 3.2, we have that

$$
\begin{aligned}
f(p) - f(p_j) &= \mathbb{E}\big[U^\#(p)\big]\big(\mathbb{E}\big[(p - C)^+\big] - \mathbb{E}\big[(p_j - C)^+\big]\big) \\
&\quad + \mathbb{E}\big[\underbrace{U^\#(p) - U^\#(p_j)}_{\leq 0}\big]\mathbb{E}\big[(p_j - C)^+\big] \\
&\quad + \big(\mathbb{E}\big[\mathbb{I}\{p_j < C \leq p\}\big]\big)\mathbb{E}\big[U^*(p)\big] \\
&\quad + \mathbb{E}\big[\mathbb{I}\{C \leq p_j\}\big]\left(\mathbb{E}\big[\underbrace{U^*(p) - U^*(p_j)}_{\leq 0}\big]\right) \\
&\leq (p - p_j) + m(p - p_j) \leq \frac{1 + m}{K} ,
\end{aligned}
$$

where in the second-to-last inequality we used Assumption 1.3, and in the last inequality we used the fact that $p - p_j \leq p_{j+1} - p_j = 1/K$.

Hence, for each $k \in \{0, 1, \ldots, K - 1\}$, by setting $\Delta_k := \max_{j \in \{0,1,\ldots,K-1\}} f(p_j) - f(p_k)$ and noticing that $\sup_{p \in [0,1]} f(p) - f(P_t) \leq 1$, we have that

$$
\begin{aligned}
\mathcal{R}_T &\leq K^2 + \sum_{t=K^2+1}^{T}\left(\sup_{p \in [0,1]} f(p) - f(P_t)\right) \\
&\leq K^2 + \frac{(1+m)T}{K} + \sum_{t=K^2+1}^{T}\left(\max_{j \in \{0,1,\ldots,K-1\}} f(p_j) - f(p_{I_t})\right) \\
&= K^2 + \frac{(1+m)T}{K} + \sum_{t=K^2+1}^{T} \Delta_{I_t} .
\end{aligned}
$$

To declutter the notation, for each $j \in \{0, 1, \ldots, K - 1\}$, define $\alpha_j := \mathbb{E}\big[U^\#(p_j)\big]$, $\beta_j := \mathbb{E}\big[\mathbb{I}\{C \leq p_j\}\big]$, $\varphi_k := \mathbb{E}\big[U^*(p_j)\big]$, and $\psi_j := \mathbb{E}\big[(p_j - C)^+\big]$. Notice that, for every $j \in \{0, 1, \ldots, K - 1\}$, we have that

$$f(p_j) = \alpha_j \cdot \psi_j + \beta_j \cdot \varphi_j .$$

Recall that for any $u \in \text{USC}$ and price $p \in [0, 1]$, $u^\#(p) \in \text{argmax}_{q \in [0,1]}\big(u(q) - p \cdot q\big)$, so we can consider the family $(U_{t,j}^\#)_{t \in \mathbb{N}, j \in \{0,1,\ldots,K-1\}}$.

Now consider the "bad" events—events in which the estimators we defined are far from the value they aim to estimate—defined, for any $\delta \in (0,1)$ and $j \in \{0, 1, \ldots, K-1\}$, by

$$\mathcal{E}_{\varphi,\delta,j} := \left| \frac{1}{K^2} \sum_{k=1}^{K} \sum_{i=j}^{K-1} U^{\#}_{k,i}(p_i) - \varphi_j \right| \geq \zeta_{K,\delta} \; ,$$

$$\mathcal{E}_{\psi,\delta,j} := \left| \frac{1}{K^2} \sum_{k=1}^{K} \sum_{i=0}^{j-1} \mathbb{I}\{C_{k,i} \leq p_i\} - \psi_j \right| \geq \zeta_{K,\delta} \; ,$$

and for any $\delta \in (0,1)$, $j \in \{0, 1, \ldots, K-1\}$, and $n \in \mathbb{N}$, by

$$\mathcal{E}_{\alpha,\delta,n,j} := \left\{ \left| \frac{1}{n} \sum_{k=1}^{n} U^{\#}_{k,j}(p_j) - \alpha_j \right| \geq \xi_{T,K,\delta}(n) \right\} \; ,$$

$$\mathcal{E}_{\beta,\delta,n,j} := \left\{ \left| \frac{1}{n} \sum_{k=1}^{n} \mathbb{I}\{C_{k,j} \leq p_j\} - \beta_j \right| \geq \xi_{T,K,\delta}(n) \right\} \; .$$

Moreover, for every $\delta \in (0,1)$, define the global "bad" event

$$\mathcal{E}_\delta := \bigcup_{j=0}^{K-1} \left( \mathcal{E}_{\varphi,\delta,j} \cup \mathcal{E}_{\psi,\delta,j} \cup \bigcup_{n=K}^{T} \left( \mathcal{E}_{\alpha,\delta,n,j} \cup \mathcal{E}_{\beta,\delta,n,j} \right) \right) \; .$$

Now, we make two crucial claims. First, for every $\delta \in (0,1)$, we have that

$$\mathbb{P}\left[ \mathcal{E}_\delta \right] \leq \delta \; . \tag{1}$$

Second, for any $\delta \in (0,1)$, when we are in the complement of the global "bad" event $\mathcal{E}_\delta$, we have that, for every $j \in \{0, 1, \ldots, K-1\}$ and every $t \in \{K^2 + 1, K^2 + 2, \ldots, T\}$,

$$f(p_j) \leq \hat{g}_{t-1}(j) \leq f(p_j) + \frac{16 \sqrt{2 \ln \left( \frac{8KT}{\delta} \right)}}{\min \left( \sqrt{N_j(t-1)}, K \right)} \; , \tag{2}$$

whenever

$$\delta \geq 8K \max \left( Te^{-K/8}, e^{-\frac{1}{2}(K/2-1)^2} \right) \tag{3}$$

and

$$\sqrt{2 \ln \left( \frac{8TK}{\delta} \right)} \geq 1 + \sqrt{2 \ln \left( \frac{8K}{\delta} \right)} \; . \tag{4}$$

To aid readability, we defer the proofs of these two claims until after the end of this proof.

Now, when we are in the complement of the global "bad" event $\mathcal{E}_\delta$, assuming that conditions 3 and 4 hold, it follows from Equation (2) that for every $t \in \{K^2 + 1, K^2 + 2, \ldots, T\}$,

$$\max_{j \in \{0,1,\ldots,K-1\}} f(p_j) \leq \max_{j \in \{0,1,\ldots,K-1\}} \hat{g}_{t-1}(j) = \hat{g}_{t-1}(I_t)$$

$$\leq f(p_{I_t}) + \frac{16 \sqrt{2 \ln \left( \frac{8KT}{\delta} \right)}}{\min \left( \sqrt{N_{I_t}(t-1)}, K \right)} \; ,$$

and consequently,

$$\Delta_{I_t} \leq \frac{16 \sqrt{2 \ln \left( \frac{8KT}{\delta} \right)}}{\min \left( \sqrt{N_{I_t}(t-1)}, K \right)} \; .$$

Thus, when we are in the complement of the global "bad" event $\mathcal{E}_\delta$, and conditions 3 and 4 hold, we have that

$$\sum_{t=K^2+1}^{T} \Delta_{I_t} = \sum_{k=0}^{K-1} \sum_{t=K^2+1}^{T} \Delta_{I_t} \mathbb{I}\{I_t = k\}$$

$$= \sum_{k=0}^{K-1} \sum_{t=K^2+1}^{T} \Delta_{I_t} \mathbb{I}\{I_t = k\} \mathbb{I}\{N_{I_t}(t-1) \geq K^2\}$$

$$+ \sum_{k=0}^{K-1} \sum_{t=K^2+1}^{T} \Delta_{I_t} \mathbb{I}\{I_t = k\} \mathbb{I}\{N_{I_t}(t-1) < K^2\}$$

$$\leq 16\sqrt{2\ln\left(\frac{8KT}{\delta}\right)} \sum_{k=0}^{K-1} \sum_{t=K^2+1}^{T} \frac{\mathbb{I}\{I_t = k\}}{K}$$

$$+ 16\sqrt{2\ln\left(\frac{8KT}{\delta}\right)} \sum_{k=0}^{K-1} \sum_{t=K^2+1}^{T} \frac{\mathbb{I}\{I_t = k\}}{\sqrt{N_{I_t}(t-1)}}$$

$$\leq 16\sqrt{2\ln\left(\frac{8KT}{\delta}\right)} \frac{T}{K} + 16\sqrt{2\ln\left(\frac{8KT}{\delta}\right)} \sum_{k=0}^{K-1} \sum_{l=K}^{N_k(T)} \frac{1}{\sqrt{l}}$$

$$\leq 16\sqrt{2\ln\left(\frac{8KT}{\delta}\right)} \frac{T}{K} + 16\sqrt{2\ln\left(\frac{8KT}{\delta}\right)} \sum_{k=0}^{K-1} 2\sqrt{N_k(T)}$$

$$\leq 16\sqrt{2\ln\left(\frac{8KT}{\delta}\right)} \left(\frac{T}{K} + 2\sqrt{KT}\right),$$

where the last inequality follows from Jensen's inequality. Hence, noticing that $\mathcal{R}_T \leq T$, when 3 and 4 hold, we conclude that

$$R_T = \mathbb{E}[\mathcal{R}_T] = \mathbb{E}[\mathcal{R}_T \mathbb{I}_{\mathcal{E}_\delta}] + \mathbb{E}[\mathcal{R}_T \mathbb{I}_{\mathcal{E}_\delta^c}]$$

$$\leq T\mathbb{P}[\mathcal{E}_\delta] + K^2 + \frac{(1+m)T}{K} + 16\sqrt{2\ln\left(\frac{8KT}{\delta}\right)} \left(\frac{T}{K} + 2\sqrt{KT}\right)$$

$$\leq T\delta + K^2 + \frac{(1+m)T}{K} + 16\sqrt{2\ln\left(\frac{8KT}{\delta}\right)} \left(\frac{T}{K} + 2\sqrt{KT}\right).$$

Now, if we set $K := \lceil T^{1/3} \rceil$ and $\delta = \frac{1}{T}$, recalling that $T \geq 5 \cdot 10^7$, we have that 3 and 4 hold, and the conclusion follows by simple estimations. $\square$

To conclude the proof of Theorem 3.3, we now present the proof of the two missing claims (1) and (2) that we used in the proof of Theorem 3.3.

*Proof of claim* (1). For any $t \in \{K^2 + 1, K^2 + 2, \dots, T\}$ and any $j \in \{0, 1, \dots, K-1\}$, we have that

$$\frac{1}{K^2} \sum_{k=1}^{K} \sum_{i=j}^{K-1} U_{k,i}^{\#}(p_i) - \varphi_j$$

$$= \frac{1}{K^2} \sum_{k=1}^{K} \sum_{i=j}^{K-1} U_{k,i}^{\#}(p_i) - \mathbb{E}\left[U^*(p_j)\right]$$

$$= \frac{1}{K} \sum_{k=1}^{K} \sum_{i=j}^{K-1} \left( \int_{p_i}^{p_{i+1}} U_{k,i}^{\#}(p_i) \, d\lambda - \mathbb{E}\left[ \int_{p_i}^{p_{i+1}} U_{k,i}^{\#}(\lambda) \, d\lambda \right] \right)$$

$$= \frac{1}{K} \sum_{k=1}^{K} \sum_{i=j}^{K-1} \left( \int_{p_i}^{p_{i+1}} U_{k,i}^{\#}(p_i) \, d\lambda - \mathbb{E}\left[ \int_{p_i}^{p_{i+1}} U_{k,i}^{\#}(p_i) \, d\lambda \right] \right)$$

$$+ \frac{1}{K} \sum_{k=1}^{K} \sum_{i=j}^{K-1} \left( \mathbb{E}\left[ \int_{p_i}^{p_{i+1}} U_{k,i}^{\#}(p_i) \, d\lambda \right] - \mathbb{E}\left[ \int_{p_i}^{p_{i+1}} U_{k,i}^{\#}(\lambda) \, d\lambda \right] \right)$$

$$= \frac{1}{K} \sum_{k=1}^{K} \sum_{i=j}^{K-1} \left( \int_{p_i}^{p_{i+1}} U_{k,i}^{\#}(p_i) \, d\lambda - \mathbb{E}\left[ \int_{p_i}^{p_{i+1}} U_{k,i}^{\#}(p_i) \, d\lambda \right] \right)$$

$$+ \mathbb{E}\left[ \sum_{i=j}^{K-1} \int_{p_i}^{p_{i+1}} \left( U^{\#}(p_i) - U^{\#}(\lambda) \right) d\lambda \right] =: (\star) .$$

Additionally, since $U^{\#}$ is monotonically non-increasing (see Lemma 3.2), using a telescopic argument, we have that

$$0 \leq \sum_{i=j}^{K-1} \int_{p_i}^{p_{i+1}} \left( U^{\#}(p_i) - U^{\#}(\lambda) \right) d\lambda$$

$$\leq \sum_{i=j}^{K-1} \int_{p_i}^{p_{i+1}} \left( U^{\#}(p_i) - U^{\#}(p_{i+1}) \right) d\lambda$$

$$= \frac{1}{K} \sum_{i=j}^{K-1} \left( U^{\#}(p_i) - U^{\#}(p_{i+1}) \right) = \frac{U^{\#}(p_j) - U^{\#}(1)}{K} \leq \frac{1}{K} .$$

It directly follows that

$$|(\star)| \leq \frac{1}{K} \sum_{k=1}^{K} \sum_{i=j}^{K-1} \left( \int_{p_i}^{p_{i+1}} U_{k,i}^{\#}(p_i) \, d\lambda - \mathbb{E}\left[ \int_{p_i}^{p_{i+1}} U_{k,i}^{\#}(p_i) \, d\lambda \right] \right) + \frac{1}{K}$$

$$= \frac{1}{K} \sum_{k=1}^{K} \sum_{i=0}^{K-1} \left( \mathbb{I}\{j \leq i\} \int_{p_i}^{p_{i+1}} U_{k,i}^{\#}(p_i) \, d\lambda - \mathbb{E}\left[ \mathbb{I}\{j \leq i\} \int_{p_i}^{p_{i+1}} U_{k,i}^{\#}(p_i) \, d\lambda \right] \right) + \frac{1}{K}$$

$$=: (\circ) + \frac{1}{K} .$$

Hence,

$$
\begin{aligned}
\mathbb{P}\left[\mathcal{E}_{\varphi,\delta,j}\right] = \mathbb{P}\left[\left|\frac{1}{K^2}\sum_{k=1}^{K}\sum_{i=j}^{K-1}U_{k,i}^{\#}(p_i) - \varphi_j\right| \geq \zeta_{K,\delta}\right] &= \mathbb{P}\left[|(\star)| \geq \frac{1+\sqrt{2\ln(8K/\delta)}}{K}\right] \\
&\leq \mathbb{P}\left[(\circ) + \frac{1}{K} \geq \frac{1}{K} + \frac{\sqrt{2\ln(8K/\delta)}}{K}\right] \\
&= \mathbb{P}\left[\sum_{k=1}^{K}\sum_{i=0}^{K-1}\left(\mathbb{I}\{j \leq i\}\int_{p_i}^{p_{i+1}}U_{k,i}^{\#}(p_i)\,\mathrm{d}\lambda\right.\right. \\
&\qquad\left.\left.- \mathbb{E}\left[\mathbb{I}\{j \leq i\}\int_{p_i}^{p_{i+1}}U_{k,i}^{\#}(p_i)\,\mathrm{d}\lambda\right]\right) \geq \sqrt{2\ln(8K/\delta)}\right].
\end{aligned}
$$

Noting that the last probability involves the sum of $K^2$ zero-mean independent $\left[-\frac{1}{K}, \frac{1}{K}\right]$-valued random variables, the previous inequality, together with Hoeffding's inequality, implies that

$$
\mathbb{P}\left[\mathcal{E}_{\varphi,\delta,j}\right] \leq 2\exp\left(\frac{-2\left(\sqrt{2\ln(8K/\delta)}\right)^2}{K^2 \cdot \frac{4}{K^2}}\right) = \frac{\delta}{4K}.
$$

Analogously, we can prove that for any $t \in \{K^2+1, K^2+2, \ldots, T\}$ and any $j \in \{0,1,\ldots,K-1\}$,

$$
\mathbb{P}\left[\mathcal{E}_{\psi,\delta,j}\right] \leq \frac{\delta}{4K}.
$$

Moreover, for any $n \in \{K, K+1, \ldots, T\}$ and any $j \in \{0,1,\ldots,K-1\}$, we have that

$$
\begin{aligned}
\mathbb{P}[\mathcal{E}_{\alpha,\delta,n,j}] = \mathbb{P}\left[\left|\frac{1}{n}\sum_{k=1}^{n}U_{k,j}^{\#}(p_j) - \alpha_j\right| \geq \xi_{T,K,\delta}(n)\right] \\
= \mathbb{P}\left[\left|\sum_{k=1}^{n}\left(U_{k,j}^{\#}(p_j) - \mathbb{E}\left[U_{k,j}^{\#}(p_j)\right]\right)\right| \geq \sqrt{2n\ln(8KT/\delta)}\right],
\end{aligned}
$$

where the last probability involves the sum of $n$ zero-mean independent $[-1,1]$-valued random variables. Thus, once again by Hoeffding's inequality, we have that

$$
\mathbb{P}[\mathcal{E}_{\alpha,\delta,n,j}] \leq 2\exp\left(\frac{-2\left(\sqrt{2n\ln(8KT/\delta)}\right)^2}{4n}\right) = \frac{\delta}{4KT}.
$$

An analogous argument, for any $n \in \{K, K+1, \ldots, T\}$ and any $j \in \{0,1,\ldots,K-1\}$, shows that

$$
\mathbb{P}[\mathcal{E}_{\beta,\delta,n,j}] \leq 2\exp\left(\frac{-2\left(\sqrt{2n\ln(8KT/\delta)}\right)^2}{4n}\right) = \frac{\delta}{4KT}.
$$

Finally, using a union bound, we obtain that

$$
\begin{aligned}
\mathbb{P}[\mathcal{E}_\delta] = \mathbb{P}\left[\bigcup_{j=0}^{K-1}\left(\mathcal{E}_{\varphi,\delta,j} \cup \mathcal{E}_{\psi,\delta,j} \cup \bigcup_{n=K}^{T}\left(\mathcal{E}_{\alpha,\delta,n,j} \cup \mathcal{E}_{\beta,\delta,n,j}\right)\right)\right] \\
\leq \sum_{j=0}^{K-1}\left(\mathbb{P}[\mathcal{E}_{\varphi,\delta,j}] + \mathbb{P}[\mathcal{E}_{\psi,\delta,j}] + \sum_{n=K}^{T}\mathbb{P}[\mathcal{E}_{\alpha,\delta,n,j}] + \sum_{n=K}^{T}\mathbb{P}[\mathcal{E}_{\beta,\delta,n,j}]\right) \\
\leq \sum_{j=0}^{K-1}\left(\frac{\delta}{4K} + \frac{\delta}{4K} + \sum_{n=K}^{T}\frac{\delta}{4KT} + \sum_{n=K}^{T}\frac{\delta}{4KT}\right) \leq \delta,
\end{aligned}
$$

which concludes the proof of claim (1). $\qquad\square$

*Proof of claim* (2). To prove claim (2), we assume from this point onward that we are in the complement of the global "bad" event $\mathcal{E}_\delta$ and that conditions 3 and 4 hold. For any $j \in \{0, 1, \ldots, K-1\}$ and any $t \in \{K^2 + 1, K^2 + 2, \ldots T\}$, we have that

$$
\begin{aligned}
f(p_j) &= \alpha_j \cdot \psi_j + \beta_j \cdot \varphi_j \\
&\leq \left( \frac{\sum_{k=1}^{N_j(t-1)} U_{k,j}^{\#}(p_j)}{N_j(t-1)} + \xi_{T,K,\delta}\big(N_j(t-1)\big) \right) \cdot \left( \frac{1}{K^2} \sum_{k=1}^{K} \sum_{i=0}^{j-1} \mathbb{I}\{C_{k,i} \leq p_i\} + \zeta_{K,\delta} \right) \\
&\quad + \left( \frac{\sum_{k=1}^{N_j(t-1)} \mathbb{I}\{C_{k,j} \leq p_j\}}{N_j(t-1)} + \xi_{T,K,\delta}\big(N_j(t-1)\big) \right) \cdot \left( \frac{1}{K^2} \sum_{k=1}^{K} \sum_{i=j}^{K-1} U_{k,i}^{\#}(p_i) + \zeta_{K,\delta} \right) \\
&= \left( \frac{A_j\big(N_j(t-1)\big)}{N_j(t-1)} + \xi_{T,K,\delta}\big(N_j(t-1)\big) \right) (G_j + \zeta_{K,\delta}) \\
&\quad + \left( \frac{B_j\big(N_j(t-1)\big)}{N_j(t-1)} + \xi_{T,K,\delta}\big(N_j(t-1)\big) \right) (F_j + \zeta_{K,\delta}) \\
&= \hat{g}_{t-1}(j) \\
&= \left( \frac{\sum_{k=1}^{N_j(t-1)} U_{k,j}^{\#}(p_j)}{N_j(t-1)} + \xi_{T,K,\delta}\big(N_j(t-1)\big) \right) \cdot \left( \frac{1}{K^2} \sum_{k=1}^{K} \sum_{i=0}^{j-1} \mathbb{I}\{C_{k,i} \leq p_i\} + \zeta_{K,\delta} \right) \\
&\quad + \left( \frac{\sum_{k=1}^{N_j(t-1)} \mathbb{I}\{C_{k,j} \leq p_j\}}{N_j(t-1)} + \xi_{T,K,\delta}(N_j(t-1)) \right) \cdot \left( \frac{1}{K^2} \sum_{k=1}^{K} \sum_{i=j}^{K-1} U_{k,i}^{\#}(p_i) + \zeta_{K,\delta} \right) \\
&\leq \big(\alpha_j + 2\xi_{T,K,\delta}\big(N_j(t-1)\big)\big) \cdot (\psi_j + 2\zeta_{K,\delta}) + \big(\beta_j + 2\xi_{T,K,\delta}\big(N_j(t-1)\big)\big) \cdot (\varphi_j + 2\zeta_{K,\delta}) \\
&\leq \alpha_j \psi_j + \beta_j \varphi_j + 8 \big(\zeta_{K,\delta} + \xi_{T,K,\delta}\big(N_j(t-1)\big)\big) = f(p_j) + 8 \big(\zeta_{K,\delta} + \xi_{T,K,\delta}\big(N_j(t-1)\big)\big) \\
&\leq f(p_j) + \frac{16\sqrt{2\ln\left(\frac{8KT}{\delta}\right)}}{\min\left(\sqrt{N_j(t-1)}, K\right)} \,,
\end{aligned}
$$

where in the second-to-last inequality, we used the fact that $\alpha_j, \beta_j, \varphi_j, \psi_j \in [0,1]$ and that $2\zeta_{K,\delta}, 2\xi_{T,K,\delta} \in [0,1]$ (by condition 3), and in the last inequality, we used Equation (4), concluding the proof of claim (2). $\qquad\square$

## D   Missing details for Section 4

**Theorem D.1** (Theorem 4.1, Restated). *Assume that Assumptions 1.1–1.4 hold. Then, the worst-case regret of any algorithm satisfies*

$$ R_T = \Omega\big(T^{2/3}\big) \,. $$

*Proof.* In Appendix A, we show how to associate every instance $(S_t, B_t)_{t\in\mathbb{N}}$ of the bilateral trade problem with a corresponding instance $(C_t, U_t)_{t\in\mathbb{N}}$ of the RMNP such that, for any price posted by the learner, the feedback and reward remain identical. Moreover, we notice that this correspondence satisfies the following equivalences:

A. $(S_t, B_t)_{t\in\mathbb{N}}$ is independent if and only if $(C_t, U_t)_{t\in\mathbb{N}}$ is independent.

B. $(S_t, B_t)_{t\in\mathbb{N}}$ is an identically distributed stochastic process if and only if $(C_t, U_t)_{t\in\mathbb{N}}$ is an identically distributed stochastic process.

C. For all $t \in \mathbb{N}$, $S_t$ and $B_t$ are independent if and only if $C_t$ and $U_t$ are independent.

D. For all $t \in \mathbb{N}$, $S_t$ has a $m$-bounded density (i.e., it has a $m$-Lipschitz cdf) if and only if $C_t$ has a $m$-bounded density (i.e., it has a $m$-Lipschitz cdf).

Consider the RMNP under Assumptions 1.1–1.4. Suppose, by contradiction, that there exists a learner's strategy $\alpha$ that achieves a regret strictly less than $cT^{2/3}$ for all instances of this problem, where $c \geq 11/672$.

Now, consider an arbitrary instance of repeated bilateral trade $(S_t, B_t)_{t \in \mathbb{N}}$, where the sequence is an i.i.d. stochastic process (iid) and, for every $t \in \mathbb{N}$, $S_t$ and $B_t$ are independent (iv) and admit densities bounded by $m \geq 24$ (bd). By the reduction established in Appendix A, there exists a corresponding instance of the RMNP, denoted by $(C_t, U_t)_{t \in \mathbb{N}}$. We observe that, by properties A and B, the sequence $(C_t, U_t)_{t \in \mathbb{N}}$ is an i.i.d. stochastic process, satisfying Assumption 1.1. Additionally, by property C, $C_t$ and $U_t$ are independent for all $t \in \mathbb{N}$, satisfying Assumption 1.2. Furthermore, property D implies that for all $t \in \mathbb{N}$, $C_t$ admits an $m$-bounded density, which in turn implies that its cdf is $m$-Lipschitz; thus, Assumption 1.3 is satisfied. Finally, by the structure of the reduction described in Appendix A, Assumption 1.4 is also satisfied:

$$\forall t \in \mathbb{N}, \; U^*(1) = (B_t - 1)\mathbb{I}\{1 \leq B_t\} = \begin{cases} (B_t - 1) \cdot 0 & \text{if } 1 > B_t \\ 0 \cdot 1 & \text{if } 1 = B_t \end{cases} = 0.$$

Thus, $(C_t, U_t)_{t \in \mathbb{N}}$ is an instance of the RMNP satisfying Assumptions 1.1–1.4, and therefore, by assumption, $\alpha$ achieves a regret of less than $cT^{2/3}$ under $(C_t, U_t)_{t \in \mathbb{N}}$.

However, since the feedback and reward under $(S_t, B_t)_{t \in \mathbb{N}}$ and under $(C_t, U_t)_{t \in \mathbb{N}}$ are identical (as shown in Appendix A), $\alpha$ must also achieve regret strictly less than $cT^{2/3}$ when run on the bilateral trade instance $(S_t, B_t)_{t \in \mathbb{N}}$. Since $(S_t, B_t)_{t \in \mathbb{N}}$ was chosen arbitrarily from the class of repeated bilateral trade instances satisfying the (iid), (iv), and (bd) assumptions, this conclusion holds for all instances in the class. This result contradicts the minimax lower bound established in [Cesa-Bianchi et al., 2021, Theorem 4.4] (see Appendix F), which states that no learner's strategy can achieve regret below $cT^{2/3}$ for all instances of the repeated bilateral trade problem satisfying these assumptions.

Thus, no learner's strategy can achieve regret strictly less than $cT^{2/3}$ across all instances of the RMNP satisfying Assumptions 1.1–1.4. Consequently, the worst-case regret of any algorithm satisfies $R_T = \Omega\left(T^{2/3}\right)$.

$\square$

# E  Missing details for Section 5

**Theorem E.1** (Theorem 5.1, Restated). *Assume that at least one of Assumptions 1.1–1.4 does not hold. Then, even if the remaining three hold, the worst-case regret of any algorithm satisfies*

$$R_T = \Omega(T).$$

*Proof.* We prove this statement by showing that relaxing any one of the four assumptions, while maintaining the other three, leads to linear regret.

**Relaxing Assumption 1.2.** We first show that any algorithm suffers linear regret when Assumption 1.2 is relaxed.[6] As detailed below, this is due to a lack-of-observability phenomenon: we construct two environments, $\mu$ and $\nu$, which are indistinguishable from the learner's perspective but have distinct optimal and (highly) suboptimal regions.

To do this, we let $\{u_x\}_{x \in [0,1]}$ be the family of utility functions defined, for every $x \in [0,1]$, by

$$u_x(q) = x\mathbb{I}\{q = 1\}.$$

Observe that each utility function $u_x$ is uniquely determined by the value $x = u_x(1)$.

---

[6]This proof is an adaption of the proof sketch of Theorem 4.5 in Cesa-Bianchi et al. [2021].

We then define the following product sets in the space of cost-utility pairs:

$$R_1 = \left[0, \frac{1}{8}\right] \times \left\{u_x : x \in \left[\frac{3}{8}, \frac{1}{2}\right]\right\},$$

$$R_2 = \left[\frac{2}{8}, \frac{3}{8}\right] \times \left\{u_x : x \in \left[\frac{7}{8}, 1\right]\right\},$$

$$R_3 = \left[\frac{4}{8}, \frac{5}{8}\right] \times \left\{u_x : x \in \left[\frac{5}{8}, \frac{6}{8}\right]\right\}.$$

along with their mirror reflections across the anti-diagonal, defined by the transformation $T$ : $[0,1] \times \{u_x\}_{x \in [0,1]} \to [0,1] \times \{u_x\}_{x \in [0,1]}, (c, u_x) \mapsto (1-x, u_{1-c})$:

$$R_4 = T(R_1), \quad R_5 = T(R_2), \quad R_6 = T(R_3).$$

We let $r_1, \ldots, r_6$ denote uniform (product) measures on $R_1, \ldots, R_6$, respectively, and characterize our two environments by

$$\boldsymbol{\mu} := \frac{1}{3}(r_1 + r_2 + r_3), \quad \boldsymbol{\nu} := \boldsymbol{\mu} \circ T = \frac{1}{3}(r_4 + r_5 + r_6).$$

We now consider a sequence of supplier costs and retailer utilities $(C, U), (C_1, U_1), (C_2, U_2), \ldots$ drawn from either $\boldsymbol{\mu}$ or $\boldsymbol{\nu}$.

When the underlying distribution is $\boldsymbol{\mu}$, the expected gain from trade $p \mapsto \mathbb{E}\big[g(p, C, U)\big]$ is uniquely maximized at $p = \frac{3}{8}$, and if $p \in \left[\frac{1}{2}, 1\right]$, then

$$\mathbb{E}\left[g\left(\frac{3}{8}, C_t, U_t\right)\right] - \mathbb{E}\big[g(p, C_t, U_t)\big] \geq \frac{1}{12}.$$

Symmetrically, when the underlying distribution is $\boldsymbol{\nu}$, the expected gain from trade is uniquely maximized at $p = \frac{5}{8}$, and if $p \in \left[0, \frac{1}{2}\right]$, then

$$\mathbb{E}\left[g\left(\frac{5}{8}, C_t, U_t\right)\right] - \mathbb{E}\big[g(p, C_t, U_t)\big] \geq \frac{1}{12}.$$

All that remains is to show that, for every price $p \in [0, 1]$, the feedback $(\mathbb{I}\{C \leq p\}, U^{\#}(p)) = (\mathbb{I}\{C \leq p\}, \mathbb{I}\{p \leq U(1)\})$ is indistinguishable under the distributions $\boldsymbol{\mu}$ and $\boldsymbol{\nu}$.

We can notice that the feedback observed indicates which block the pair $(C, U)$ lies in:

$$B_{00}(p) = \{(c, u) : c > p, \ u(1) < p\}, \qquad B_{01}(p) = \{(c, u) : c > p, \ u(1) \geq p\},$$
$$B_{10}(p) = \{(c, u) : c \leq p, \ u(1) < p\}, \qquad B_{11}(p) = \{(c, u) : c \leq p, \ u(1) \geq p\}.$$

By direct verification,

$$T(B_{00}(1-p)) = B_{00}(p), \qquad T(B_{01}(1-p)) = B_{10}(p),$$
$$T(B_{10}(1-p)) = B_{01}(p), \qquad T(B_{11}(1-p)) = B_{11}(p).$$

Hence, for any $(i, j) \in \{0, 1\}^2$,

$$\boldsymbol{\nu}\big(B_{ij}(p)\big) = \boldsymbol{\mu}\Big(T^{-1}\big(B_{ij}(p)\big)\Big) = \boldsymbol{\mu}\big(B_{ji}(1-p)\big) = \boldsymbol{\mu}\big(B_{ij}(p)\big),$$

where the last equality follows by the symmetry of the rectangles. Consequently, the distribution of the feedback is identical under both $\boldsymbol{\mu}$ and $\boldsymbol{\nu}$.

Thus, any learner's strategy must perform sub-optimally in at least half of the rounds within one of the two indistinguishable worlds. Since the loss suffered in each sub-optimal round is at least $\frac{1}{12}$, the expected per-round regret is at least $\frac{1}{2} \cdot \frac{1}{12} = \frac{1}{24}$.

Therefore, we conclude that the regret of any learner's strategy is at least $\frac{1}{24}T$.

**Relaxing Assumption 1.1.** We now adapt the above construction to show that linear regret is inevitable when Assumption 1.1 is relaxed.[7] Specifically, we introduce two families of oblivious adversarial environments, $\mathcal{F}_{\boldsymbol{\mu}}$ and $\mathcal{F}_{\boldsymbol{\nu}}$, each generating sequences of independent (but not necessarily identically distributed) cost-utility pairs. We then show that the worst-case regret suffered by any learner's strategy under environments from $\mathcal{F}_{\boldsymbol{\mu}}$ or $\mathcal{F}_{\boldsymbol{\nu}}$ is at least as large as the regret suffered under $\boldsymbol{\mu}$ or $\boldsymbol{\nu}$, respectively.

Recalling that, for every $i = 1, \ldots, 6$, the distribution $\boldsymbol{r}_i$ is defined as the uniform measure on the product set $R_i$, we now construct the two adversarial families:

$$
\mathcal{F}_{\boldsymbol{\mu}} := \left\{ \bigotimes_{t=1}^{T} \boldsymbol{r}_{i_t} : (i_t)_{t \in [T]} \in \{1, 2, 3\}^T \right\}, \quad \mathcal{F}_{\boldsymbol{\nu}} := \left\{ \bigotimes_{t=1}^{T} \boldsymbol{r}_{i_t} : (i_t)_{t \in [T]} \in \{4, 5, 6\}^T \right\}.
$$

Now, for any learner's strategy, the regret suffered against the worst adversary in $\mathcal{F}_{\boldsymbol{\mu}}$ is lower bounded by the (expected) regret the learner suffers when the adversary is selected uniformly at random from the family $\mathcal{F}_{\boldsymbol{\mu}}$. Choosing this adversary uniformly at random from $\mathcal{F}_{\boldsymbol{\mu}}$ is equivalent to independently and uniformly selecting each index $i_t$ from $\{1, 2, 3\}$. This yields a sequence of cost-utility pairs $(C_1, U_1), (C_2, U_2), \ldots$ whose marginal distribution at each round is precisely $\boldsymbol{\mu}$.

Hence, the maximum regret any learner's strategy suffers from an adversary in $\mathcal{F}_{\boldsymbol{\mu}}$ is at least as large as the regret suffered under the environment $\boldsymbol{\mu}$.

Analogously, one can show that an adversary chosen uniformly at random from $\mathcal{F}_{\boldsymbol{\nu}}$ is distributed according to $\boldsymbol{\nu}$. Thus, the maximum regret any learner's strategy suffers from an adversary in $\mathcal{F}_{\boldsymbol{\nu}}$ is at least as large as the regret suffered under the environment $\boldsymbol{\nu}$.

Combining these results, we can conclude that the regret of any learner's strategy is at least $\frac{1}{24}T$.

**Relaxing Assumption 1.3.** To show linear regret when lifting Assumption 1.3, we construct an infinite family of i.i.d. environments,

$$
\{\boldsymbol{\mu}_x\}_{x \in [\frac{2}{5}, \frac{3}{5}]},
$$

and show that any learner's strategy suffers linear regret under at least one environment in this family.[8]

To do this, for every $x \in \left[\frac{2}{5}, \frac{3}{5}\right]$, we define two $[0, 1]$–valued utility functions

$$
u_x(q) = x\mathbb{I}\{q = 1\}, \qquad u_1(q) = \mathbb{I}\{q = 1\}.
$$

Then, letting $\delta_x$ be the Dirac measure on $x$, we set

$$
\boldsymbol{\mu}_x := \left( \tfrac{1}{2}\delta_0 + \tfrac{1}{2}\delta_x \right) \otimes \left( \tfrac{1}{2}\delta_{u_x} + \tfrac{1}{2}\delta_{u_1} \right).
$$

Under the environment $\boldsymbol{\mu}_x$, the expected gain from trade is uniquely maximized at $p = x$, and if $p \neq x$, we have that

$$
\mathbb{E}\big[g(x, C, U)\big] - \mathbb{E}\big[g(p, C, U)\big] = \frac{\min\{x, 1-x\}}{4} \geq \frac{\frac{2}{5}}{4} = \frac{1}{10}, \qquad (\star)
$$

where the inequality follows from the fact that $x \in \left[\frac{2}{5}, \frac{3}{5}\right]$.

Since this environment is stochastic, without any loss of generality, it is enough to show that deterministic algorithms suffer linear regret. Fix an arbitrary deterministic algorithm and a time horizon $T \in \mathbb{N}$. Let $p_1, \ldots, p_T$ denote the sequence of posted prices. Since the interval $\left[\frac{2}{5}, \frac{3}{5}\right]$ is uncountable, there exists some

$$
x' \in \left[\tfrac{2}{5}, \tfrac{3}{5}\right] \setminus \{p_1, \ldots, p_T\}.
$$

Under the environment $\boldsymbol{\mu}_{x'}$, the price $x'$ is never posted. Thus, for any sequence of supplier costs and retailer utilities $(C_1, U_1), (C_2, U_2), \ldots$ drawn from $\boldsymbol{\mu}_{x'}$, inequality $(\star)$ holds at every round $t \in [T]$, implying that

$$
\mathbb{E}\big[g(x', C_t, U_t)\big] - \mathbb{E}\big[g(p_t, C_t, U_t)\big] \geq \frac{1}{10}.
$$

Summing over $t = 1, \ldots, T$ yields a regret of at least $\frac{1}{10}T$.

---

[7]This proof is an adaption of the proof of Theorem 3 in Cesa-Bianchi et al. [2024].

[8]This proof is an adaption of the proof of Theorem 4.6 in Cesa-Bianchi et al. [2021].

**Relaxing Assumption 1.4.**  Finally, we show that no learner's strategy can achieve sublinear worst-case regret without prior knowledge of $\mathbb{E}[U^*(a)]$ for some price $a \in [0, 1]$, even when the first three assumptions 1.1–1.3 still hold.

Define the two functions

$$f \colon [0, 1] \to [0, 1],$$
$$q \mapsto -(q - 0.7)^2 + 0.5,$$
$$h \colon [0, 1] \to [0, 1],$$
$$q \mapsto -(q - 0.7)^2 + 0.65.$$

Then, letting $\lambda$ be the Lebesgue measure on $[0, 1]$, we define

$$\boldsymbol{\mu} := \lambda \otimes \delta_f, \quad \boldsymbol{\nu} := \lambda \otimes \delta_h.$$

Suppose that the sequence of supplier costs and retailer utility functions $(C, U)$, $(C_1, U_1)$, $(C_2, U_2), \ldots$ is drawn i.i.d. from $\boldsymbol{\mu}$ or $\boldsymbol{\nu}$.

When the underlying distribution is $\boldsymbol{\mu}$, the expected gain from trade $p \mapsto \mathbb{E}\big[g(p, C, U)\big]$ is uniquely maximized at $p = \frac{5}{7}$, and if $p \in \left[\frac{3}{4}, 1\right]$, then

$$\mathbb{E}\left[g\left(\frac{5}{7}, C_t, U_t\right)\right] - \mathbb{E}\big[g(p, C_t, U_t)\big] \geq \frac{1}{2500}.$$

On the other hand, when the underlying distribution is $\boldsymbol{\nu}$, the expected gain from trade is uniquely maximized at $p = \frac{13}{14}$, and if $p \in [0, \frac{3}{4}]$, then

$$\mathbb{E}\left[g\left(\frac{13}{14}, C_t, U_t\right)\right] - \mathbb{E}\big[g(p, C_t, U_t)\big] \geq \frac{3}{4000}.$$

Under both distributions $\boldsymbol{\mu}$ and $\boldsymbol{\nu}$, we have $U^\#(p) = 0.7 - \frac{p}{2}$ for every price $p \in [0, 1]$. Hence, the feedback $\big(\mathbb{I}\{C \leq p\}, U^\#(p)\big)$ is identical under both distributions, making the two cases indistinguishable to the learner.

Thus, any learner's strategy must perform sub-optimally in at least half of the rounds within one of the two indistinguishable worlds. Since the loss suffered in each sub-optimal round is at least $\min\left(\frac{1}{2500}, \frac{3}{4000}\right)$, the expected per-round regret is at least $\frac{1}{2} \cdot \min\left(\frac{1}{2500}, \frac{3}{4000}\right) = \frac{1}{5000}$.

Therefore, we conclude that the regret of any learner's strategy is at least $\frac{1}{5000}T$.

$\square$

# F    Other results

The following is a direct consequence of Corollary 4 of Milgrom and Segal [2002].

**Theorem F.1** (Envelope Theorem, Milgrom and Segal 2002)**.** *Assume that $f \colon [0, 1]^2 \to [0, 1]$ is such that*

- *for all $p \in [0, 1]$, we have that the function $q \mapsto f(q, p)$ is upper semi-continuous.*

- *for all $q \in [0, 1]$, we have that the function $(q, p) \mapsto \partial_p f(q, p)$ is continuous.*

*then, if we define*

$$V \colon [0, 1] \to [0, 1], \qquad p \mapsto \max_{q \in [0, 1]} f(q, p)$$

*and $x^\circ \colon [0, 1] \to [0, 1]$ is any selection such that for any $p \in [0, 1]$,*

$$x^\circ(p) \in \operatorname*{argmax}_{q \in [0, 1]} f(q, p)$$

*we have that $x^\circ$ is integrable, and, for any $p \in [0, 1]$,*

$$V(p) = V(0) + \int_0^p x^\circ(\lambda)\, \mathrm{d}\lambda.$$

For completeness, we also report the freezing lemma (see, e.g., Cesari and Colomboni [2021]) that we use in the proof of Theorem 3.3 to convert pseudo-regret guarantees into regret guarantees.

**Lemma F.2** (The freezing lemma). *Let $(\Omega, \mathcal{F}, \mathbb{P})$ be a probability space. Let $(\mathcal{V}, \mathcal{F}_{\mathcal{V}})$ and $(\mathcal{W}, \mathcal{F}_{\mathcal{W}})$ be two measurable spaces. Let $f : \mathcal{V} \times \mathcal{W} \to [0, +\infty]$, $V : \Omega \to \mathcal{V}$, $W : \Omega \to \mathcal{W}$ be three measurable functions. If $V$ and $W$ are $\mathbb{P}$-independent, then*

$$\mathbb{E}[f(V, W) \mid V] = [\mathbb{E}[f(v, W)]]_{v=V}$$

$\mathbb{P}$-*almost surely, where the right-hand side is the composition*

$$[\mathbb{E}[f(v, W)]]_{v=V} = (v \mapsto \mathbb{E}[f(v, W)]) \circ V.$$

Moreover, we report the repeated bilateral trade regret lower bound theorem [Cesa-Bianchi et al., 2021, Theorem 4.4] that we use in the proof of Theorem 4.1 to establish the optimality of Algorithm 1 for the RMNP.

**Theorem F.3** (Theorem 4.1, Cesa-Bianchi et al. 2021). *In the realistic-feedback model, for all horizons T, the minimax regret $R_T^\star$ satisfies*

$$R_T^\star := \inf_\alpha \sup_{(S,B) \sim D} R_T(\alpha) \geq cT^{2/3},$$

*where $c \geq 11/672$, the infimum is over all of the learner's strategies $\alpha$, and the supremum is over all distributions of the seller $S$ and buyer $B$ such that:*

(iid)  *$(S_1, B_1), (S_2, B_2), \cdots \sim (S, B)$ is an i.i.d. sequence;*

(iv)  *$S$ and $B$ are independent of each other;*

(bd)  *$S$ and $B$ admit densities bounded by $m \geq 24$.*

