# OpenReview forum: "Online Learning in the Repeated Mediated Newsvendor Problem"
_NeurIPS.cc/2025/Conference — NeurIPS 2025 poster_

### Official Review · Reviewer_LmM4 · 2025-06-26

**Clarity:** 2
**Significance:** 3
**Originality:** 4
**Rating:** 4
**Confidence:** 4

**Summary:**

This paper considers the problem of online learning with a mediator learning the behavior of the supplier and the buyer. Specifically, the learner proposes a price $p_t$ at each round $t$, observes $1\\{p_t\leq c_t\\}$ indicating whether the supplier accepts the trade and $U_t^{\sharp}(p_t)$, indicating the amount the buyer will buy based on the buyer's utility function $U_t$. The goal is to maximize the total gain-from-trade. The author shows that this is a strict generalization of online bilateral trading problem, a problem setting that is well motivated and recently studied broadly.

Under the assumptions that $c_t$ and $U_t$ are independent and stochastic, the authors first provide a UCB-based algorithm achieving $\tilde{O}(T^{2/3})$ regret, together with a nearly matching lower bound. The algorithm works on the discretized price set and is split into two phases. The first phase is a pure exploration phase, which aims to provide a UCB estimate for $\mathbb{E}[(p-c)^+]$ and $\mathbb{E}[U^*(p)]$ that is fixed in later rounds. The second phase then uses the previously obtained UCB estimates and further applies a UCB based algorithm based on the estimated gain-from-trade for each discretized price. The authors also show that all the assumptions they made are necessary in the sense that if one of these assumptions does not hold, there is no algorithm that can achieve sublinear regret. This is proven by reducing this problem to online bilateral trading. The authors also conduct experiments to verify their theoretical findings.

**Questions:**

I have some questions about the derivations:
- Assumption 1.3 is not well-defined. What is the Lipschitz constant that is assumed for the CDF for $C$?
- In Line 895, can the authors explain more about how they use Assumption 1.3? It seems that the definition of $m$ is missing.
- As for the UCB estimates in the first phase, I wonder whethere there are some intuitions that the confidence interval is $O(1/K)$ for **each** price $p$ given only $K$ samples for that price? My (vague) intuition is that the UCB estimates for smaller $p$ requires less samples to approximate and for larger prices, they can utilize the observations from smaller prices since the estimate is based on CDF but I am not clear about this. Can the authors explain more about this?
- Also, can the authors comment more about how more relaxed feedbacks could potentially improve the regret rate?

**Ethical Concerns:**

["NO or VERY MINOR ethics concerns only"]

**Final Justification:**

Based on the rebuttal and other reviewers' review, I keep my original rating of this paper.

**Limitations:**

Yes.

**Paper Formatting Concerns:**

None.

**Quality:**

4

**Strengths And Weaknesses:**

Strengths:
- The problem is well-motivated and is a generalization of the online bilateral trading problem.
- The authors provide an almost optimal upper and lower bound for this problem under the assumptions they made. Moreover, the authors show that the assumptions are necessary to achieve sublinear regret for this problem.
- I checked the proofs and they look correct in general, except for some parts that I will point out in later sections.
- The author also provide empirical evidence to show the effectiveness of their algorithm, together with the justifications on the assumptions they made.

Weakness: I do not find major issues for this paper but I think the derivations in this paper can be further polished.
- The Lipschitz constant is not defined in Assumption 1.3.
- The explanation of the confidence interval \mathcal{E}_{\psi,\delta,j} and \mathcal{E}_{\phi,\delta,j} is missing. Specifically, I am not sure whether $U_{k,i}^{\sharp}(p)$ (shown in line 903) is defined. In addition, there are some inconsistancy in the proof. For example, in line 903, $\mathcal{E}_{\phi,\delta,j}$ is defined with respect to $\frac{1}{K(K-j)}$ and in line 924, it changes to $\frac{1}{K^2}$. While I understand the high-level intuition of the proof and most of the results are as expected, I do not understand some detailed derivations (see questions) and I wonder whether the authors can explain more in detail.

---

> ### Author Rebuttal · Authors · 2025-07-30
>
> **Is $U^{\sharp}_{k,i}(p)$ defined?**
> We define the operator $\sharp$ that maps any USC function $u$ into its transformed $u^{\sharp}$ on line 52 of the main part of the paper, at the beginning of Section 1.2 (Formal setting).
> But the reviewer raises a good point that adding a reminder around line 903 would improve clarity, as the reader may have forgotten its definition by that point.
> We implemented this change for the revised version.
>
> **Typos on line 903.**
> Thank you for catching it!
> The terms $1/K(K-j)$ and $1/Kj$ on line 903 are indeed typos and should both be $1/K^2$, consistently with the rest of the paper.
> We double-checked and didn't find any other occurrences of these typos. We updated the revised version with this change.
>
> **Assumption 1.3 has a missing "$m$".**
> Thank you for catching this typo as well!
> Assumption 1.3 should indeed be "*The cumulative distribution function (cdf) of the supplier’s cost $C$ is $m$-Lipschitz*". In Line 895, we use precisely this assumption when we upper bound
> $\mathbb E [ \mathbb I$ { $p_j < C \le p$ } $]=\mathbb P [ p_j < C \le p ] \le m \cdot (p - p_j)$.
> We updated the revised version with this change.
>
> **Why few samples give $O(1/K)$-sized confidence intervals.**
> The reviewer's intuition is roughly correct.
> For each price $p_j$, we have $K(K-j+1)$ samples for the term that involves $F_j$, and $K j$ samples for the term that involves $G_j$.
> As the reviewer's intuition suggested, for lower prices $p_j$, the terms $G_j$ are easier to estimate (since we are estimating the integral of a $[0,1]$-valued function in $[0,p_j]$), while for higher prices the terms $F_j$ are easier to estimate (since we are estimating the integral of a $[0,1]$-valued function in $[p_j,1]$).
> The trick to prove this intuition (see lines 927-930) is to apply Hoeffding's inequality to a slightly larger set of random variables, generated by adding *identically-zero* random variables until reaching a total of $K^2$ zero-mean, $[-\frac1K,\frac1K]$ independent random variables.
>
> As a sanity check, notice that when $j = 1$ (resp., $j = K$), we are in fact using $K^2$ samples to obtain the order $1/K$ in the confidence interval for $F_1$ (resp. $G_K$).
>
> **Could the authors comment more about how more relaxed feedback models could potentially improve the regret rate?**
> Intriguing question.
> A feedback of theoretical interest that is commonly considered is "full feedback".
> In our setting, this would mean that $C_t$ and $U_t$ are revealed at the end of each round.
> We make the following conjectures for this setting:
> 1) Independence of $C_t$ and $U_t$ is not necessary to guarantee learnability, and neither is Lipschitnzess of the cdf of $C_t$;
> 2) A follow-the-leader (FTL) approach should lead to a $\sqrt{T}$ regret guarantee. However, we suspect that some novel uniform convergence argument would be needed to obtain the $\sqrt{T}$ bound for FTL due to the specific complexities of the problem.
>
> ---
>
> We hope our answers helped clarify all lingering doubts.
>
> If this is the case, we'd kindly ask the reviewer if they would consider readjusting their score.

---

> ### Comment · Reviewer_LmM4 · 2025-08-05
>
> I thank the authors for their detailed response. I think most of my points have been addressed. Given that, I will maintain my current score and discuss further with other reviewers in later phases.

---

> > ### Author Response · Authors · 2025-08-06
> >
> > We thank the reviewer once again for their thoughtful engagement. If any remaining concerns stand in the way of a more positive assessment, we remain open and eager to address them.

---

### Official Review · Reviewer_QQVf · 2025-06-27

**Clarity:** 3
**Significance:** 3
**Originality:** 3
**Rating:** 4
**Confidence:** 2

**Summary:**

The paper studies a repeated mediated newsvendor problem (RMNP) where the learner is a mediator that facilitates trades between suppliers and retailers in a sequence of interactions. It proposes an online mediator’s pricing strategy that achieves the optimal regret rate with theoretical guarantees and experimental validation.

**Questions:**

1. While the paper assumes retailer utility functions belong to the class of upper semi-continuous (USC) functions, is this assumption common in prior newsvendor literature?

2. Does the model account for the overage cost of unsold goods. If not, how does this cost impact the derived regret bounds?

**Ethical Concerns:**

["NO or VERY MINOR ethics concerns only"]

**Final Justification:**

The authors have added repeated experiments and included error bars, which improve the completeness of the results. However, the model still relies on somewhat strong assumptions, which limit its practicality and broader applicability. Therefore, despite these improvements, I have decided to keep my original score.

**Limitations:**

Overall, the paper does not show significant limitations. Some minor limitations include:

1. The experimental section lacks error bars and repeated trials, which hinders the evaluation of algorithm stability under stochastic variations.

2. The theoretical framework relies on idealized assumptions, which may not hold in real-world supply chains with non-stationary dynamics or concentrated cost distributions. Such gaps could limit the model’s applicability.

**Paper Formatting Concerns:**

No major formatting issues were noticed.

**Quality:**

3

**Strengths And Weaknesses:**

**Strengths:**

**Strong theoretical contributions:** The derived regret bound is tight and matches the lower bounds up to some logarithmic factors. The authors also verify the necessity of all assumptions involved in the study.

**Clear structure:** This paper is well-organized with intuitive proof sketches.

**Weaknesses:**

The paper's theoretical assumptions exhibit notable gaps in aligning with real-world supply chain dynamics, particularly regarding Assumption 1.1 (i.i.d. sequences). While the proof of assumption necessity in Appendix E constructs counterexamples to show linear regret, it overlooks weaker relaxations that better model practical scenarios. For example, adaptive algorithms might still achieve sublinear regret in mildly dependent processes (e.g., ARIMA-simulated time-series with seasonal trends), where temporal correlations are weak but non-negligible.

In the experiment part, the study omits repeated trials across diverse supplier cost distributions and retailer utility functions, failing to report error bars or confidence intervals for regret. This absence prevents robust assessment of the algorithm’s stability under stochastic fluctuations.

---

> ### Author Rebuttal · Authors · 2025-07-30
>
> **Is the USC assumption common?**
> Good question. As far as we know, it is not a common assumption.
> To the best of our knowledge, the most common assumption by far is that utilities are (at least) Lipschitz.
> Our USC assumption is weaker (in fact, even strictly weaker than continuity alone) but still sufficient for our theory.
> In particular, all (possibly discontinuous) right-continuous non-decreasing functions are upper semi-continuous and can therefore be encompassed by our theory.
> We will highlight this in the revised version.
>
> **Does the model account for the average cost of unsold goods? If not, how does this cost impact the derived regret bounds?**
> In our setting, goods are produced only after they are ordered.
> For instance, in online freelance marketplaces, freelancers only complete tasks, i.e. produce the "goods", after both the freelancer and the user agree to the job.
> Similarly, in ridesharing platforms, drivers only incur fuel/car maintenance costs during the ride, which occurs after an agreement is made between the driver and the rider.
> However, this is an interesting direction to explore for future work and we thank the reviewer for their suggestion.
>
> **Add error bars.**
> We thank the reviewer for the suggestion. Please see our response to Question 3 of reviewer m1KJ (labeled by: **Add error bar and statistical significance.**) for a detailed explanation of how the updated plots will look like in the revised version.
>
> **Are our assumptions realistic?**
> We thank the reviewer for raising this point and are happy to clarify.
>
> As shown in Theorem 5.1, if any one of the assumptions is relaxed, sublinear regret is no longer achievable.
> Thus, under non-stationary dynamics or concentrated cost distributions (which violate Assumptions 1.1 and 1.3, respectively), the problem becomes unlearnable, regardless of the model.
>
>
> That being said, an interesting direction for future work would be to explore relaxations of the problem setting.
> For example, the weaker benchmark of $\alpha$-regret may allow for learnability under non-stationary dynamics.
>
> We also note that several real-world applications in which our assumptions are approximately satisfied are discussed in the motivations section.
> Please refer to our response to Question 1 of reviewer m1KJ (labeled by: **Can you discuss your assumptions further?**) for a more in-depth explanation of the real-world relevance of these assumptions.
>
> ---
>
> We hope our answers helped clarify all lingering doubts.
>
> If this is the case, we'd kindly ask the reviewer if they would consider readjusting their score.

---

> > ### Comment · Reviewer_QQVf · 2025-08-05
> >
> > Thank you for your rebuttal—it addressed most of my concerns and questions. However, I still have some concerns regarding the assumptions. While the counterexample in the proof shows that sublinear regret cannot be achieved without each of the assumption, it may not fully justify the necessity of the assumption in its current form. It remains unclear whether a weaker version of the assumption might still suffice. Could you provide more insights into why your proof can rule out the possibility of achieving sublinear regret under any weaker assumption?

---

> > > ### Author Response · Authors · 2025-08-06
> > >
> > > Thank you for this insightful and subtle question.
> > > It touches on a core issue in the logic of impossibility results.
> > >
> > > To clarify: please note that we do not make the claim that our set of assumptions is the only way to ensure sublinear regret, nor that the negation of any one of them implies that the problem becomes unlearnable.
> > > Rather, our impossibility results show that if one drops any one of the assumptions *without replacing it with anything else*, then sublinear regret cannot be guaranteed in general.
> > >
> > > The reviewer's suspicion is correct, and learnability *may* be preserved under alternative or strictly weaker assumptions.
> > > For example, one could weaken Assumption 1.1 to require that the sequence
> > > $(C_2, U_2), (C_3, U_3), \dots$ (starting from round 2 onward) is i.i.d.
> > > This is formally weaker than Assumption 1.1 but still suffices for our positive results.
> > > Likewise, we do not rule out the existence of orthogonal assumptions that also permit sublinear regret.
> > >
> > > We want to conclude by emphasizing the inherent trade-off between modeling realism and theoretical tractability. Our assumptions aim to hit a sweet spot: capturing a broad class of sufficiently realistic scenarios (see again our response to m1KJ) while retaining enough structure to enable strong theoretical guarantees.

---

### Official Review · Reviewer_m1KJ · 2025-07-02

**Clarity:** 3
**Significance:** 2
**Originality:** 3
**Rating:** 4
**Confidence:** 3

**Summary:**

This paper studies the repeated newsvendor problem, a game with three players: a supplier, a retailer, and a mediator. Different from literature where the supplier determines the trading price, in this paper, the mediator determines the price to facilitate trades and maximize the social welfare over time. The authors introduce an online learning algorithm for this mediator. Their research offers robust theoretical findings, including highly effective learning rates and matching lower bounds. Additionally, the paper thoroughly examines why certain initial conditions (key assumptions) are essential for the problem's learnability.

**Questions:**

The mediator's objective is to maximize social welfare. In real-world mediated platforms, other objectives might exist (e.g., maximizing mediator's profit, ensuring fairness between participants). How might changing the mediator's objective function affect the learnability, regret bounds, or even the mechanism design properties?

**Ethical Concerns:**

["NO or VERY MINOR ethics concerns only"]

**Final Justification:**

The authors satisfactorily addressed all of my questions. My rating remains unchanged.

**Limitations:**

yes

**Quality:**

2

**Strengths And Weaknesses:**

On the positive side, this paper considers a learning approach to maximize the sum of the net gains of the supplier and the retailer during repeated interactions. I found the problem presented by the paper interesting and relevant. In general, this paper proposed an innovative research topic.

On the other hand, although the necessity of assumptions is technically justified by Theorem 5.1, a deeper intuitive explanation of their real-world meaning would make more sense. For example, what practical situations might violate these assumptions, and how often do such scenarios occur?

The claim in Theorem 2.1 is not surprising, especially given the i.i.d. assumption for each round's supplier and retailer. Under this assumption, participants naturally have no reason to act dishonestly.

The paper does not report error bars or other measures of statistical significance for its experimental results. While the experiments illustrate the theoretical findings, providing some indication of variability or confidence intervals (e.g., shaded error bars in Figure 1) would improve the credibility of the empirical results and align with standard reporting guidelines.

Finally, the paper states that providing information on computer resources for experiments is "unnecessary" because the experiments are "small-scale." However, for better reproducibility and transparency, it is good practice to include basic details about the computing environment (e.g., CPU type, RAM) and typical execution times, even for small experiments.

---

> ### Author Rebuttal · Authors · 2025-07-30
>
> **Can you discuss your assumptions further?**
> Yes, we would be happy to provide a deeper explanation.
>
> In a real-world sense, Assumption 1.1 (i.i.d. supplier costs and retailer utility functions) implies that the behaviors of suppliers/retailers follow the same general pattern over time without being influenced by past interactions.
> In well-known platforms, where the number of users is high, the platform rarely sees the same supplier/retailer within a short window and the population of suppliers/retailers tends to be stable over time.
> This makes the i.i.d. assumption a reasonable approximation.
> For example, Airbnb, one of our motivating applications, has over 5 million hosts (i.e., suppliers) as of March 2025.
> On the other hand, small-scale platforms may violate the assumption as the same participants may return frequently, and the population may shift over time.
> Therefore, if the focus is on established platforms, the i.i.d. model is appropriate.
>
> For Assumption 1.2 (independence between the supplier's cost and retailer's utility function), we refer the reviewer to our response to Question 1 of reviewer LSey (labeled by: **Can we assume suppliers' costs are independent of retailers' utilities?**) for a detailed explanation.
>
> Regarding Assumption 1.3 (Lipschitz supplier cost cdf), this condition ensures that the cost of supplying a good varies gradually across suppliers, rather than abruptly.
> In other words, there are no large clusters of suppliers with costs concentrated in a narrow price range.
> This is a natural assumption guaranteeing that small changes in the trading price lead to stable changes in the likelihood of the trade.
> As a counterexample, suppose that a large proportion of rideshare drivers in a population refuel at the same gas station in a region where fuel prices are relatively stable.
> In this case, many drivers have nearly identical costs, resulting in a sharp spike in the cdf of supplier costs around the shared fuel price, which is a violation of our assumption.
> As a result, a small change in the trading price could cause a large proportion of drivers to accept a trade, causing unstable or unpredictable outcomes.
> We would argue that these types of scenarios with highly concentrated distributions are not overly common in practice.
> Thus, the assumption of Lipschitzness is well-justified in most real-world scenarios.
>
> Assumption 1.4 (the expectation of the retailer’s optimal utility at price 1 is 0) means that, on average, retailers gain nothing if the trading price is set to its maximum value.
> Intuitively, the highest possible price is too high to justify any purchase by a retailer, resulting in zero net gain.
> For example, a rideshare rider may open the Uber app to check if the fare is below their personal threshold that they are willing to pay.
> If the price is at its highest possible value, it will exceed their threshold, making it likely the user will either walk or switch to a different service, such as Lyft.
> This behavior corresponds to a capped utility function, which is a common family of retailer utility functions where utility is zero beyond a certain price.
> One practical scenario in which this assumption is violated is when demand is inelastic: for instance, if a retailer must buy a certain quantity of the good no matter the cost.
> Following our previous example, the rider may be willing to purchase a ride at the maximum price if no alternative transportation options exist, e.g. if Uber is the only available service.
> That being said, such cases typically require a pure monopoly over a market or an emergency setting, which are relatively rare.
>
> **In an i.i.d. setting, the incentive-compatibility proved in Theorem 2.1 is not that surprising.**
> We will clarify what follows in the revised version.
> Note that, even in an i.i.d. setting, participants *may* have an incentive to misreport their evaluations. Consider, for example, first-price auctions. Even if a bidder (identified by their valuation) is drawn i.i.d. from a family of bidders and will never return after a single interaction, they still have an incentive to shade their bids (i.e., to under-report their evaluations), because that is the only way they can gain a positive utility in a first-price auction mechanism.
> This is due to how the mechanism operates more than how the valuations of the participants are generated.
>
> **Add error bar and statistical significance.**
> Thank you for the suggestion. We have updated the code for generating the plots to include error bars for the regret (95\% confidence intervals calculated using the normal approximation). We would be happy to include these updated plots in the revised version.
>
> Unfortunately, we are not allowed to post any images here. However, to provide some context for the updates, the following table shows the algorithm's average regret and 95\% confidence intervals at every 100,000 iterations, where the algorithm was run for 700,000 iterations across 100 repeated trials (with a uniform supplier cost distribution and linear-capped retailer utility):
>
> | Iteration | Mean Regret | 95% CI |
> |---:|---:|---:|
> | 100,000 | 3,041 | ± 19 |
> | 200,000 | 4,871 | ± 31 |
> | 300,000 | 6,271 | ± 43 |
> | 400,000 | 7,431 | ± 59 |
> | 500,000 | 8,434 | ± 70 |
> | 600,000 | 9,321 | ± 84 |
> | 700,000 | 10,133 | ± 98 |
>
>
> To give a visual, we note that although the error bands vary slightly across different supplier cost distributions and retailer utility functions, their thickness in both directions generally matches or is less than that of the regret curves (when run with 100 repeated trials). In other words, when the error bands are included, each regret curve visually appears, at most, about three times thicker.
>
> **Add info about your computing environment and execution times.**
> We would be happy to include these details in the revised version. For reference, we ran our experiments on a 14-inch MacBook Pro with an M1 Pro chip ($10$-core CPU, 16-core GPU) and 16 GB of RAM. With this configuration, each plot takes at most around 12 minutes to generate when running each experiment for 100 repeated trials using the updated code.
>
> **How might changing the mediator's objective function affect the learnability, regret bounds, or even the mechanism design properties?**
> Great question! The short answer is that those properties will vary depending on the objective.
> Consider the following three examples to illustrate some possible variations.
>
> 1.  If the learner earns a commission $\alpha \in [0,1]$ (thought of as a percentage) and tries to maximize their return, their objective function becomes $p\mapsto\alpha p U^{\sharp}_t(p) \mathbb{I}$ { $C_t \le p$ }.
>
> 2. If the learner posts two different prices $(p,q)\in [0,1]^2$ with $p \le q$ to the supplier/retailer and tries to maximize their margin, their objective function becomes $ (p,q) \mapsto (q-p) U^{\sharp}_t(p) \mathbb I$ { $C_t \le p$ }.
>
> 3. If the learner tries to maximize a fair notion of gain from trade, their objective function, inspired by Bachoc et al. [2024], could become $ p \mapsto \min$ { $U^*_t(p), p- C_t$ } $\cdot \mathbb I$ { $C_t \le p$ }.
>
> Regarding learnability/regret bounds:
>
> - In the first two examples, the learner has access to bandit feedback.
> Hence, under suitable assumptions that guarantee the (at least one-sided-)Lipschitzness of the objective function, one could obtain regret rates of order $T^{2/3}$ in the first case (because the space is $1$-dimensional) and $T^{3/4}$ in the second case (because the pair of prices $(p,q)$ such that $p \le q$ form a $2$-dimensional manifold).
>
> - For the third example, the learner cannot observe their own reward (the received feedback $\mathbb{I}$ { $C\le p$ } and $U^{\sharp}(p)$ is not enough to directly reconstruct $(C-p)$ or $U^{*}(p)$, and hence the learner does not have bandit feedback). Therefore, the problem becomes more challenging and some additional tricks would be needed to be able to reconstruct enough of the expected reward to guarantee learnability.
>
> Regarding the mechanism-design properties:
>
> - In the first and third examples, the utilities of both the supplier and the retailer stay the same, so Theorem 2.1 should extend naturally.
>
> - In the second example, a proof like the one in Theorem 2.1 shows that incentive-compatibility and individual rationality are preserved.
> However, the mechanism can only be guaranteed to be *weakly* budget balanced (meaning that the learner does not subsidize trades but is allowed to siphon value from them).
>
> We are happy to add these future directions to Section 7 in the revised version of this work.
>
> ---
>
> We hope our answers helped clarify all lingering doubts.
>
> If this is the case, we'd kindly ask the reviewer if they would consider readjusting their score.

---

> > ### Comment · Reviewer_m1KJ · 2025-08-06
> >
> > Thank you for the clarification, I have no further questions.

---

> ### Author Response · Authors · 2025-08-06
>
> We thank the reviewer for their thoughtful engagement. If any remaining concerns stand in the way of a more positive assessment, we would be grateful for the opportunity to address them.

---

### Official Review · Reviewer_LSey · 2025-07-03

**Clarity:** 1
**Significance:** 2
**Originality:** 3
**Rating:** 4
**Confidence:** 3

**Summary:**

The paper studies an online learning version of the repeated mediated newsvendor problem (RMNP), where a platform (mediator) sets trading prices between suppliers and retailers over time with the goal of maximizing social welfare (gain from trade). At each round, a supplier with a private production cost and a retailer with a private utility function arrive, and the platform proposes a price. The supplier decides whether to accept the price, and if so, the retailer decides how much to purchase. The authors propose a no-regret algorithm that achieves an $\widetilde{O}(T^{2/3})$ regret rate under four key assumptions: i.i.d. supply and demand, independent supplier and retailer types, Lipschitz supplier cost distribution, and bounded retailer utility at the highest price. They also provide a matching lower bound showing this rate is unimprovable up to log factors, and impossibility results demonstrating that relaxing any of the assumptions leads to linear regret. The paper positions RMNP as a generalization of bilateral trade and provides empirical simulations validating the theoretical results.

**Questions:**

1) In motivating Assumption 1.1, the paper states: "The first two assumptions simply mean that this is a stochastic i.i.d. setting in which the supplier’s production cost does not directly influence the retailer’s utility function. This models the real-life fact that the utility earned by the retailer is only directly affected by the trading price."

Could the authors clarify the basis for this assertion? What evidence or reasoning supports the assumption that these are independent in realistic settings, beyond analytical convenience?

2) The algorithm forms an overall UCB for gain-from-trade by multiplying UCBs for acceptance probability and surplus given trade. Could the authors clarify whether this product construction maintains the desired confidence level, or whether more direct concentration for the product would be needed to ensure statistical validity?

3) The algorithm relies on a uniform discretization of the price space and spends $T^{2/3}$ periods on pure exploration. Could the authors comment on why they chose this non-adaptive exploration strategy, and whether more efficient alternatives could achieve similar regret guarantees with improved practical efficiency?

**Ethical Concerns:**

["NO or VERY MINOR ethics concerns only"]

**Final Justification:**

The authors have provided technically sound responses that helped clarify my main concerns. In particular, their explanation of the independence assumption, the construction of confidence bounds for gain-from-trade, and the necessity of pure exploration addressed the key technical and modeling issues I had raised. While the assumption of independence still limits applicability, the authors have acknowledged this and suggested a possible contextual extension. That said, I still find the novelty of the work to be moderate, as it builds closely on prior bilateral trade literature. On balance, I am updating my evaluation to a borderline accept.

**Limitations:**

Limitations of the work are not properly discussed in the paper.

**Paper Formatting Concerns:**

No formatting concerns.

**Quality:**

2

**Strengths And Weaknesses:**

Strengths:

1) The paper provides substantial theoretical results, including a no-regret algorithm with regret guarantees that match a proven lower bound up to logarithmic factors. The impossibility results clearly delineate the boundary between settings where sublinear regret is achievable and where it is not.

2) The formulation of RMNP and its connection to bilateral trade mechanisms represent a novel contribution

Weaknesses:

1) The assumption of independence between supplier cost and retailer utility are quite strong and limit the applicability of the results to real-world mediated markets, where these characteristics are often dependent through shared context. This creates a contradiction with the Motivation section.

2) The problem setting is not clearly described. The formal presentation is difficult to follow, with notation and decomposition choices.

3) The proposed algorithm spends $T^{2/3}$ periods for pure exploration of the pricing grid which is not practical.

---

> ### Author Rebuttal · Authors · 2025-07-30
>
> **Can we assume suppliers' costs are independent of retailers' utilities?**
> Fair question. A first reason why the assumption is reasonable is related to the distinct economic drivers of suppliers and retailers.
> Supply‑side costs typically depend on factors like input prices, technology, scale, and logistics, whereas demand‑side utilities depend on consumer preferences, substitution patterns, and timing.
> These factors do not always overlap. For example, a fall in steel prices lowers bicycle-frame costs but does not necessarily change a commuter's willingness to pay for the bike.
>
> Another reason is that in many mediated markets, the item traded is fixed (e.g., a specific SKU, ride, or room‑night). In this case, switching a supplier for another affects the production cost but not the retailer's valuation of that identical item.
> In other words, if it is always the same type of item that's traded, it is realistic to assume that the random draw of the supplier would not influence that of the retailer.
>
> A more technical reason has to do with necessity (not mere convenience) for learnability.
> Theorem 5.1 shows that allowing any statistical dependence between $C$ and $U$ makes sublinear regret unachievable: the optimal benchmark itself becomes unidentifiable.
>
> That said, if markets exhibit shared shocks (e.g., seasonal effects) that influence both $C_t$ and $U_t$, it could be natural to extend our proposed setting to a *contextual* setting where a context vector $x_t$ is available to the learner at the beginning of each round.
> This is an intriguing direction that, to the best of our knowledge, has only very recently been studied in simpler mediated settings (see, e.g., *A Tight Regret Analysis of Non-Parametric Repeated Contextual Brokerage*, ICML 2025).
> We conjecture that, due to the contextual nature of this setting, different techniques would be needed to address this extension.
> We will add a brief discussion of this extension to the revised version, as well as the clarifications above, so that the concrete motivations behind the independence assumption become clearer for future readers.
>
> **UCB and confidence levels.**
> We confirm that the product of the UCBs the reviewer mentions does constitute a UCB for our objective, but the confidence level of the product differs from those of the factors.
> In our work, we choose specific confidence levels for the factors $\zeta_{K,\delta}$ and $\xi_{T,K,\delta}(n)$ defined at the bottom of page 5 so that an appropriately carried out derivation leads to the desired bounds.
> The full derivation can be found in the appendix (it ends at the bottom of page 26).
>
> **Necessity of pure exploration.**
> Great question!
> We conjecture that some amount of "pure exploration" is indeed necessary, and that the amount allocated by our algorithm is likely the correct order of magnitude.
> The intuition is that our objective function (the expected gain from trade) can be decomposed as a sum of products of terms, some of which are not directly observable (see Step 2 of the proof sketch on pages 6-7).
> To estimate these hidden terms (which can be thought of as integrals over prices in $[0,1]$), one has to post prices that are spread over the entire unit interval.
> This is true even if one were given two distinct prices (one of which is optimal) and asked solely to determine which of the two was the optimal one.
> Still, the reviewer might wonder if perhaps the decomposition we provide is not the "best" that can be proven, and if the objective can be written in a different way that includes no hidden integral terms.
> Again, we believe this is not possible because of a "revealing action" phenomenon that was observed in the seminal work of Cesa-Bianchi et al. (*A Regret Analysis of Bilateral Trade*, EC 2021), which studies a special case of our setting.
> There (see Figure 2 in the arxiv version of the paper), the authors show that the learner can be forced to post prices that are known to be suboptimal and far from potentially optimal prices in order to gather usable information about these potentially optimal prices.
> This essentially amounts to forcing the learner to spend $\Theta(T^{2/3})$ rounds of "pure exploration".
> We are happy to add this discussion to the revised version as well.
>
> ---
>
> We hope our answers helped clarify all lingering doubts.
>
> If this is the case, we'd kindly ask the reviewer if they would consider readjusting their score.

---

> > ### Comment · Reviewer_LSey · 2025-08-01
> >
> > Thank you for the clarifications.

---

> > > ### Author Response · Authors · 2025-08-06
> > >
> > > We thank the reviewer for their thoughtful engagement. If any remaining concerns stand in the way of a positive assessment, we would be grateful for the opportunity to address them.

---

### Decision · Program_Chairs · 2025-09-17

**Decision:**

Accept (poster)

**Comment:**

This paper studies an online learning version of the repeated mediated newsvendor problem. On the upside,
- the reviewers acknowledge that the paper provides substantial theoretical results
- in the rebuttal, explanations of the assumptions, the construction of confidence bounds for gain-from-trade, and the necessity of pure exploration, detailed experimental results were all deemed useful

On the downside, the reviewers think that:
- the work still relies on somewhat strong modeling assumptions, which limit its practicality and broader applicability.
- the novelty may be moderate, given that the paper builds closely on prior bilateral trade literature

Given these considerations, this paper is at the borderline of acceptance.